# Improving Autoregressive Video Modeling with History Understanding

**Wenyang Luo**[1,2*†]   **Haina Qin**[3,4†]   **Bing Li**[1,2]   **Jiwen Lu**[4]   **Xin Tao**[3]   **Pengfei Wan**[3]
**Kun Gai**[3]
[1]Institute of Automation, Chinese Academy of Sciences
[2]School of Artificial Intelligence, University of Chinese Academy of Sciences
[3]Kling Team, Kuaishou Technology   [4]Tsinghua University

## Abstract

Video autoregressive generation (VideoAR) sequentially predicts future frames conditioned on history frames. Despite the advance of recent diffusion-based VideoAR, the role of conditioning signal—internal representations of history frames—remains underexplored. Inspired by the success of strong condition representations in text-conditioned generation, we investigate: *Can better internal representations of history frames improve VideoAR performance?* Through systematic analysis, we show that history representation quality positively correlates with VideoAR, and that enhancing these representations provides gains that cannot be achieved by refining future frames representations alone. Based on these insights, we propose **MiMo** (Masked History Modeling), a novel framework that seamlessly integrates representation learning into diffusion-based VideoAR. MiMo applies masks to history frame tokens and trains the model to predict masked tokens of current and future frames alongside the diffusion objective, yielding predictive and robust history representations without relying on vision foundation models (VFMs) or heavy architectural changes. Extensive experiments demonstrate that MiMo achieves competitive performance in video prediction and generation tasks while substantially improving training efficiency. Our work underscores the importance of history representations in VideoAR.

## 1 Introduction

Video autoregressive generation (VideoAR) predicts future frames conditioned on previously observed or generated frames (the history). The *history-to-future* generation process naturally aligns with the causal structure of video dynamics and enables variable-length generation (Villegas et al., 2022; Yin et al., 2025; Teng et al., 2025). However, early AR approaches (Yan et al., 2021; Hong et al., 2022; Ge et al., 2022; Villegas et al., 2022) significantly underperformed non-AR methods (Brooks et al., 2024; Ho et al., 2022; He et al., 2022b; Guo et al., 2024), primarily due to the difficulty of modeling the complex conditional distribution of future frames given history. Recently, diffusion-based VideoAR (Kondratyuk et al., 2023; Chen et al., 2024a; Song et al., 2025; Gu et al., 2025) has emerged as a promising solution, as it can approximate complex conditionals via iterative denoising of future frames from random noise, conditioned on the history frames.

Despite this progress, the conditioning signal—the representation of the history frames—remains underexplored. In text-to-image/video (T2I/T2V) and class-conditioned generation, stronger condition representations consistently improve generation quality (Esser et al., 2024; Gao et al., 2024; Kong et al., 2024; Hu et al., 2024b; Wu et al., 2025), which raises a natural question: *Can better internal representations of history enhance VideoAR performance?* [1] Intuitively, if the model's internal representations of history effectively capture the semantics and dynamics of the history frames, predicting coherent future frames should become easier. However, in current diffusion-based VideoAR, history representations are mainly learned via the diffusion objective, which may

---

*Work conducted during the author's internship at Kling Team, Kuaishou Technology. †Equal contribution.

[1]We focus on internal representations of *clean* history frames as conditions, distinct from methods that improve representations of *noisy* data within the diffusion process (Yu et al., 2024; Zhang et al., 2025).

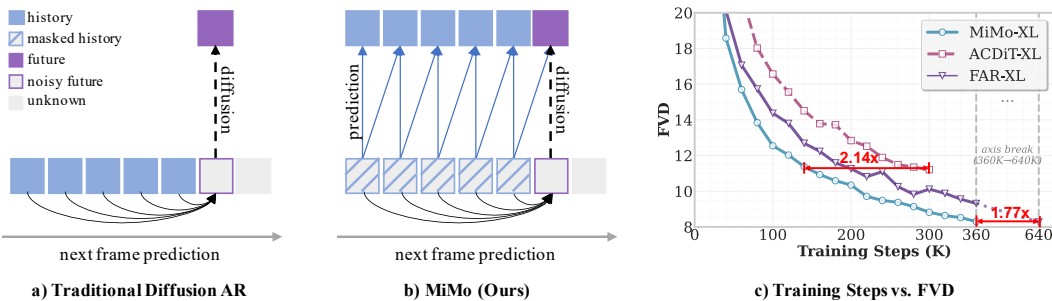

Figure 1: Good representations of history frames improve VideoAR. Our framework, MiMo, incorporates masked modeling into the history frames during training. MiMo achieves significantly faster convergence than baselines *without* using VFM.

not be optimal for learning semantically aligned, predictive condition representations. Moreover, good representations do not naturally emerge from VideoAR training, because predicting future frames requires modeling the low-level details of the future, which can hinder representation learning (Yu et al., 2024). This limitation motivates us to design dedicated learning objectives to enhance history representations and improve VideoAR performance. Importantly, we aim to achieve this without relying on external vision foundation models (VFMs) that incur substantial training costs and may suffer from out-of-distribution issues when applied to new video domains.

In this work, we demonstrate that improving history frame representations can indeed enhance VideoAR performance—an improvement that cannot be achieved by refining noisy future frame representations alone. Based on this insight, we propose *Masked History Modeling* (MiMo), a novel diffusion-based VideoAR framework *without vision foundation model (VFM)*, illustrated in Figure 1. MiMo naturally integrates masked modeling (Devlin et al., 2019; He et al., 2022a; Tong et al., 2022), a simple yet effective representation learning technique, into history frame modeling for VideoAR. Our approach works as follows: We first address the train-test discrepancy in recent methods (Chen et al., 2024a; Song et al., 2025) by incorporating clean (noise-free) history frames alongside the noisy future frames being denoised during training, similar to the approaches of Zhou et al. (2025); Hu et al. (2024a). Then, we mask (drop) portions of the history frame tokens and train the model to reconstruct the masked tokens of current and future frames in parallel with the diffusion loss. This dual objective encourages the model to learn robust history representations that help the model to predict future frames, while also improving its robustness to perturbations in history.

Unlike previous work that applies masked modeling to noisy inputs of diffusion models (Gao et al., 2023; Wei et al., 2023), which harms diffusion and requires complicated techniques to mitigate the negative effects, our approach operates on clean history frames. Our approach greatly alleviates interference with future prediction and requires minimal architectural modifications. MiMo substantially improves training efficiency and generation quality through self-supervised visual representation learning and achieves strong VideoAR performance, all without external pretrained VFMs (Yu et al., 2024; Zhang et al., 2025). In MiMo, history frames serve dual purposes: as conditions for the denoising of the future frame, and as input for self-supervised representation learning. By unifying history representation learning with future frame diffusion modeling, our framework enables high-quality representations that boost video prediction and generation.

Our main contributions are:

1. We investigate how history frame representations impact VideoAR performance and demonstrate that better representations lead to improved generation quality.

2. We propose MiMo, a simple yet effective VFM-free framework that seamlessly unifies diffusion-based VideoAR with self-supervised history representation learning.

3. Our framework demonstrates competitive video prediction and generation performance in VideoAR, achieving state-of-the-art (SOTA) results on several benchmarks.

## 2 PRELIMINARIES

**VideoAR** Given a video $\mathbf{x} = \{x_i \in \mathbb{R}^{H \times W \times 3} | i = 1, \ldots, T\}$ with $T$ frames of height $H$ and width $W$, AR approaches model the temporal sequence by generating future frames *sequentially* conditioned on historical frames, following the natural causal structure of video dynamics. VideoAR can be formulated by conditional probabilities:

$$p(x_{t+1}|x_{1:t}) = p(\text{future frame}|\text{history frames}), \tag{1}$$

where $x_{1:t} = \{x_1, x_2, \ldots, x_t\}$ represents the history frames and $x_{t+1}$ is the next frame to generate.

**Diffusion-based VideoAR** The conditional probabilities defined by Equation (1) are usually complex, which can be modeled by diffusion models (Sohl-Dickstein et al., 2015; Ho et al., 2020). Diffusion-based AR approaches models Equation (1) by learning to denoise the Gaussian-nosied future frame: $x_{t+1}^{(\tau)} = \alpha_\tau x_{t+1} + \sigma_\tau \epsilon$, conditioned on history frames $x_{1:t}$, where $\epsilon \sim \mathcal{N}(\mathbf{0}, \mathbf{I})$ and $\tau \in [0, 1]$ is noise level, and $\{\alpha_\tau, \sigma_\tau\}_\tau$ is noise schedule. This is typically done by estimating the score function $s_\theta(x_{t+1}^{(\tau)}; \tau, x_{1:t}) \approx \nabla \log p_\tau(x_{t+1}^{(\tau)}|x_{1:t})$ (Vincent, 2011). In practice, $s_\theta$ is often parameterized in alternative forms, such as v-prediction (Salimans & Ho, 2022): $v_\theta(x_{t+1}^{(\tau)}; \tau, x_{1:t}) \approx \alpha_\tau \epsilon - \sigma_\tau x_{t+1}$.

During training, diffusion forcing (Song et al., 2025; Gu et al., 2025) learns $v_\theta$ that conditions on noisy history frames $x_{1:t}^{(\tau_{1:t})} = \{x_1^{(\tau_1)}, x_2^{(\tau_2)}, \ldots, x_t^{(\tau_2)}\}$ with independent noise levels $\tau_{1:t}$,

$$\mathcal{L} = \mathbb{E}_{t, \tau_{1:t+1}, \epsilon_{t+1}, \mathbf{x}} \left[ \|\alpha_{\tau_{t+1}} \epsilon_{t+1} - \sigma_{\tau_{t+1}} x_{t+1} - v_\theta(x_{t+1}^{(\tau_{t+1})}; \tau_{1:t+1}, x_{1:t}^{(\tau_{1:t})})\|_2^2 \right]. \tag{2}$$

In contrast, complete teacher forcing (CTF) (Hu et al., 2024a; Zhou et al., 2025) eliminates train-test discrepency of diffusion forcing by conditioning on clean history frames $x_{1:t}$:

$$\mathcal{L} = \mathbb{E}_{t, \tau_{t+1}, \epsilon_{t+1}, \mathbf{x}} \left[ \|\alpha_{\tau_{t+1}} \epsilon_{t+1} - \sigma_{\tau_{t+1}} x_{t+1} - v_\theta(x_{t+1}^{(\tau_{t+1})}; \tau_{t+1}, x_{1:t})\|_2^2 \right] \tag{3}$$

During generation, the model iteratively denoises $x_{t+1}^{(\tau)}$ using the learned denoising network, starting from pure noise and gradually recovering the clean future frame. Once $x_{t+1}$ is fully denoised, it is appended to history frames for generating the subsequent frame $x_{t+2}$.

## 3 METHOD

### 3.1 OVERVIEW

We hypothesize that good history frame representations benefit VideoAR. To investigate this hypothesis, we first analyze the relationship between history frame representation quality and VideoAR performance (Section 3.2). Our findings reveal that improving history representations improves performance, and such improvement cannot be achieved by solely refining noisy future frames.

These findings motivate our approach. Additionally, we aim to avoid using VFM (Yu et al., 2024), as they may perform poorly for out-of-distribution (OOD) data, and adapting or pretraining VFMs on OOD data requires additional effort and increases complexity. Specifically, we propose **M**asked **Hi**story **Mo**deling (MiMo), a unified framework that jointly optimizes history frame representation learning and VideoAR within a single training process (Section 3.3) *without* using VFM. The insight of MiMo lies in treating history frames as noise-free conditioning signals during both training and inference, while introducing auxiliary masked video modeling objectives specifically targeting history frames. The dual objectives ensure that the model develops robust history frame representations while maintaining strong generative capabilities. MiMo can also be extended to other pretraining objectives (Oquab et al., 2023; Assran et al., 2023; Jiang et al., 2025; Wang & He, 2025), which we leave for future work.

### 3.2 EXPLORING REPRESENTATIONS OF HISTORY

In this section, we analyze the impact of history frame representations on VideoAR performance. We aim to understand whether good representations of history frames correlate with better generation quality, and whether this is necessary—in other words, whether we can achieve all benefits by

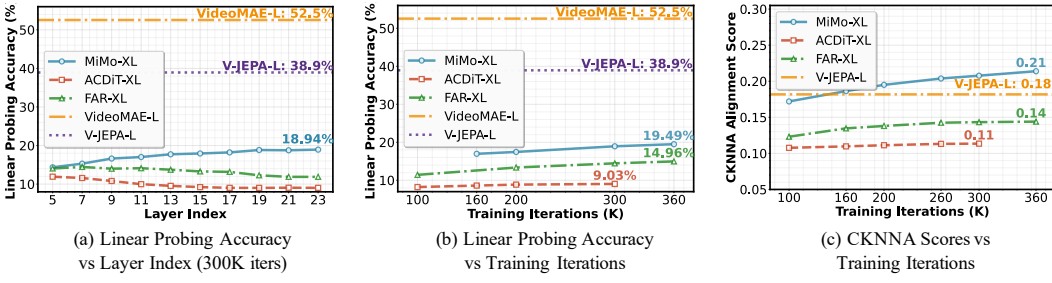

Figure 2: Exploring representations of history frames.

solely improving the representations of the noisy future frames being denoised (Yu et al., 2024). To exclude the influence of first-frame generation quality and focus on understanding the role of history frames, we conduct experiments on the K600 video prediction task, where the video prediction model predicts eleven future frames based on five given context frames. To study representation quality, we perform linear probing on K600 and measure CKNNA (Huh et al., 2024) to assess the similarity between model internal representations and pretrained representations (Yu et al., 2024). We select VideoMAE (Tong et al., 2022) and V-JEPA (Bardes et al., 2024) as VFM. All models use DFoT VAE (Song et al., 2025), with identical hyperparameters across all experiments and history guidance (see Appendix D.5) with scale 1.05 (Song et al., 2025) during inference. Details about evaluations are provided in Appendix F.

**History frame representation quality correlates with video prediction performance**  We empirically investigate the relationship between history frame representation quality and video prediction performance using the models trained as shown in Figure 1, with results summarized in Figure 2. Our main findings include: (a) History frame quality positively correlates with video prediction performance—better models exhibit higher linear probing accuracy and better alignment with VFM (measured by CKNNA). (b) During training, history frame representation quality gradually improves but consistently maintains a significant gap with pretrained models. (c) Our proposed MiMo method effectively improves history frame representation quality. Notably, MiMo changes the layer where linear probing accuracy peaks, as our method introduces decoders in later layers to execute the masked history modeling objective (see Section 3.3).

**Improving history frame representations is a feasible way to improve video prediction performance.**  We investigate whether improving history frame representation quality can enhance video prediction performance by training ACDiT-B models (Hu et al., 2024a), which take clean history frames and noisy future frames as input during training, where both history and future frames can only attend to themselves and history frames. This architecture allows us to explicitly separate the representations of history frames. We compare two approaches: one

Table 1: Improving representations of history frames.

| Method | | FVD↓ | Acc.(%)↑ |
|---|---|---|---|
| ACDiT-B | | 54.814 | 6.21 |
| REPA | History | 40.022 | 16.96 |
| | Future | 40.253 | 17.04 |
| | Both | 36.542 | 19.23 |

similar to REPA (Yu et al., 2024), which distills features from VFM into history frame representations; another introduces the MAE objective (He et al., 2022a) in history frames. Table 1 shows that both REPA and self-supervised methods can improve representation quality and subsequently enhance video prediction performance, demonstrating that improving history frame representations is feasible.

**Improving noisy future frame representations cannot replace the role of improving history frame representations.**  Besides history frame representations, the representation quality of noisy future frames also affects diffusion model generation performance (Yu et al., 2024; Zhang et al., 2025). A meaningful question is: Is it sufficient to only improve the representation quality of noisy future frames? Our answer is *no*. We train ACDiT-B models and compare introducing REPA ob-

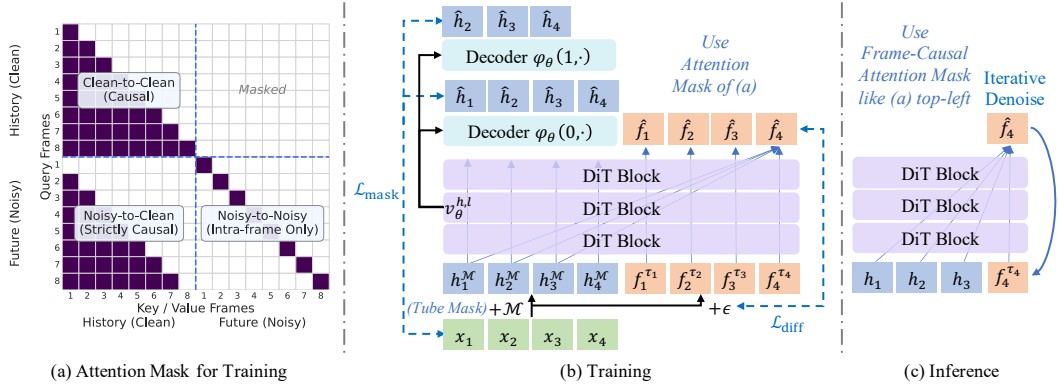

Figure 3: Framework of MiMo. (a) Attention mask used for training. Eight frames are shown. The clean history frames and noisy future frames are allowed to attend to themselves and previous history frames. (b) Training. Four frames are shown. The video clip $\mathbf{x} = \{x_1, \ldots, x_3\}$ is both used as history frames $\mathbf{h}$ and masked with a random tube mask $\mathcal{M}$, and as future frames $\mathbf{f}$ and noised with Gaussian noise $\epsilon$. The prediction targets of masked history modeling are the current and next frames. (c) AR Inference. Three history frames are already generated or provided by the user; the fourth frame is being denoised. After denoising, the fourth frame is appended to the history frames.

jectives in clean history frames, noisy future frames, or both. Table 1 demonstrates that merely improving noisy future frame representations is insufficient. Simultaneously improving both history and future frames yields benefits beyond just improving noisy future frame representations, indicating that history frames contain unique semantics. Note that our attempt to introduce MAE objective in noisy future frames (similar to Wei et al. (2023); Gao et al. (2023)) fails to surpass the performance of the ACDiT baseline without modifying the model's macro-architecture (also reported by Gao et al. (2023)). We leave such exploration for future work.

## 3.3    MiMo: Masked History Modeling

Motivated by our findings in Section 3.2, we propose MiMo to improve history representations in diffusion-based VideoAR.

**Framework Design**  The core design principle of MiMo is to leverage history frames for dual purposes: (1) as conditions for diffusion-based future frame generation, and (2) as input for self-supervised representation learning through masked modeling. This dual utilization enables the model to develop robust history frame representations that are specifically tailored for video modeling tasks. The design is shown in Figure 3.

During training, MiMo follows CTF (Hu et al., 2024a; Zhou et al., 2025), which exposes clean history frames for representation learning. Given a video clip $\mathbf{x} = \{x_1, x_2, \ldots, x_T\}$, we duplicate it as both history frames $\mathbf{h} = \{h_1, h_2, \ldots, h_T\}$ and target future frames $\mathbf{f} = \{f_1, f_2, \ldots, f_T\}$, where $\mathbf{h} = \mathbf{f} = \mathbf{x}$. The history frame $h_t$ is input without noise; it can attend to itself and all its previous history frames $h_{t' \leq t}$. Future frame $f_t$ is independently noised with Gaussian noise $\epsilon_t$ as in DFoT (Chen et al., 2024a; Song et al., 2025); it can attend to itself and all the previous future frames $h_{t' < t}$. This can be implemented by an attention mask as depicted in Figure 3(a).

The diffusion objective for future frame generation is:

$$\mathcal{L}_{\text{diff}} = \mathbb{E}_{t, \tau, \epsilon_t, \mathbf{x}, \mathcal{M}} \left[ \|\alpha_\tau \epsilon_t - \sigma_\tau f_t - v_\theta(f_t^{(\tau)}; \tau, h_{1:t}^{\mathcal{M}})\|_2^2 \right], \tag{4}$$

where $x_t^{(\tau)}$ is the noisy version of the future frame $x_t$ at diffusion timestep $\tau$, $v_\theta$ is the denoising network (v-prediction (Salimans & Ho, 2022)) conditioned on masked history frames $h_{1:t}^{\mathcal{M}}$, and $\mathcal{M}$ is a random tube mask (Tong et al., 2022) applied on the history frames with a ratio $r$ for masked history modeling (introduced below).

**Masked History Modeling**    To enhance the model's understanding of history frames, we introduce a masked modeling objective on the clean history frames. We randomly mask a subset of tokens in the history frame $h_t$ and train the model to reconstruct the masked content. *Crucially*, the reconstruction target can be either the tokens of the current history frame or any of the clean future frames $h_{t' \geq t}$. This distinguishes it from the normal diffusion objective as it allows greater flexibility.

Formally, we introduce the reconstruction loss on the masked history frames $h_{1:t}^{\mathcal{M}}$, the reconstruction target is a set of frames $\mathcal{T}_t = \{t, t+1\}$ which contain both $t$ and its next frame $t+1$:

$$\mathcal{L}_{\text{mask}} = \mathbb{E}_{t,\tau,\epsilon_t,\mathbf{x},\mathcal{M}} \left[ \frac{1}{|\mathcal{T}_t|} \sum_{t' \in \mathcal{T}_t} \|h_{t'} - \varphi_\theta(t' - t, v_\theta^{h,l}(f_t^{(\tau)}; \tau, h_{1:t}^{\mathcal{M}}))\|_2^2 \right], \tag{5}$$

where $\varphi_\theta$ is a lightweight decoder that predicts the masked tokens of frame $t' \in \mathcal{T}$, and $v_\theta^{h,l}$ is the denoising network's output features of the $l$-th layer for the history frames $h_{1:t}$.

The unified training objective combines both losses:

$$\mathcal{L}_{\text{total}} = \mathcal{L}_{\text{diff}} + \lambda \mathcal{L}_{\text{mask}}, \tag{6}$$

where the hyperparameter $\lambda$ balances the masked modeling objective.

**Inference**    During inference, as shown in Figure 3, MiMo discards decoder $\varphi_\theta$ and operates in standard AR fashion with KV cache (Zhou et al., 2025; Hu et al., 2024a; Gu et al., 2025): given clean history frames $h_{1:t-1}$, the model generates the next future frame $f_t$ through iterative denoising. The learned history representations enhance the model's ability to maintain temporal coherence and generate high-quality future content. The framework naturally supports variable-length generation by iteratively updating the history context with newly generated frames.

**Discussion**    Compared with masked diffusion that applies a masked modeling objective to denoising input (Gao et al., 2023; Wei et al., 2023), our approach operates on clean history frames and mitigates the interference with the diffusion denoising process. Thus, MiMo requires no special architectural designs that masked diffusion approaches require. Zhou et al. (2025) also corrupt history frames, but their motivation is to improve robustness to noise in history, and they apply no reconstruction target to the history frames. Thus, they are still limited in history representations. Our approach also reduces the computational costs compared with those of Zhou et al. (2025); Hu et al. (2024a) due to masking.

## 4  EXPERIMENTS

### 4.1  SETUP

**Tasks and Datasets**    We evaluate MiMo on three video modeling tasks: video prediction, unconditional video generation, and class-conditional video generation. For video prediction, we use the Kinetics-600 dataset (Carreira et al., 2018), which consists of 480,000 videos with 600 categories (class labels are *not* used). Five frames are provided as initial conditions to predict the next eleven frames. For video generation, we use the UCF-101 dataset (Soomro et al., 2012) with 13,320 videos across 101 categories. No initial frame is provided, and the model generates 16 frames.

**Implementation Details**    The architecture is based on DiT (Peebles & Xie, 2023). Our modifications are: 1) using QK normalization (Henry et al., 2020) to stabilize training, 2) incorporating RoPE (Su et al., 2024), and 3) using separate LayerNorm (Ba et al., 2016) for clean history frames[2]. The decoder is a stack of four DiT blocks with the same configuration as the model. Hyperparameter $\lambda = 0.5$. The learning rate is $8 \times 10^{-4}$ for Kinetics and $4 \times 10^{-4}$ for UCF-101, both decayed to $10^{-5}$ with cosine schedule. The global batch size is 256 for Kinetics and 128 for UCF-101. The weight decay is 0.001, and the betas for AdamW (Loshchilov & Hutter, 2017) are $(0.9, 0.99)$. The model is trained for 360K steps on Kinetics and 180K on UCF-101. For Kinetics, we use DFoT's VAE (Song et al., 2025) with a compression ratio of $4 \times 8 \times 8$ and sample 17 frames per clip with resolution

---

[2]These modifications moderately affect performance, as shown in Section 4.3.

Table 2: System comparison on Kinetics and UCF-101 with video prediction, unconditional video generation, and conditional video generation tasks. [†]: Different from the original work, we reimplemented DFoT using a causal architecture to align with the standard AR practice.

| Method | Type | Kinetics (Pred.) FVD↓ | UCF-101 (Uncond.) FVD↓ | UCF-101 (Cond.) FVD↓ |
|---|---|---|---|---|
| LVDM (He et al., 2022b) | Non-AR | – | 372 | – |
| MAGVIT (Yu et al., 2023a) | | 9.9 | – | 76 |
| MAGVITv2 (Yu et al., 2023b) | | 4.3 | – | 58 |
| Latte (Ma et al., 2024b) | | – | 478 | – |
| TATS (Ge et al., 2022) | AR | – | 420 | 332 |
| Phenaki (Villegas et al., 2022) | | 36.4 | – | – |
| Omni (Wang et al., 2024) | | 32.9 | – | 191 |
| DFoT-XL[†] (Song et al., 2025) | | 11.1 | – | – |
| ACDiT-XL (Hu et al., 2024a) | | – | – | 111 |
| MAGI-XL (Zhou et al., 2025) | | 11.5 | 298 | – |
| FAR-XL (Gu et al., 2025) | | – | 279 | 108 |
| VAE Reconstruction | AR | 3.7 | 15 | 15 |
| **MiMo-XL** | | **8.3** | **240** | **98** |

128. For UCF-101, we use FAR's (Gu et al., 2025) per-frame DC-AE (Chen et al., 2024b) with a compression ratio of $32 \times 32$ and sample 16 frames per clip with resolution 256. See Appendix D for more details.

**Inference and Evaluation** We follow Song et al. (2025) for evaluation on Kinetics, generating 50,000 random videos and computing the Fréchet Video Distance (FVD) (Karras et al., 2019) on all frames (including conditioning and generated frames) with the groundtruth videos, both resized to $64 \times 64$. On UCF-101, following FAR, we randomly sample 2,048 videos and compute the FVD against groundtruth videos, resized to $256 \times 256$.

## 4.2 MAIN RESULTS

Table 2 presents a comprehensive comparison of MiMo against state-of-the-art non-AR and AR methods across three video modeling tasks. For reference, we also report the reconstruction FVD of the VAE, which represents the upper bound of performance achievable given the groundtruth.

**Video Prediction (Pred.)** On the challenging Kinetics-600 video prediction benchmark, MiMo demonstrates exceptional performance with an FVD score of 8.3, establishing a new state-of-the-art among AR models. This represents a substantial improvement over previous AR methods, with our approach significantly outperforming DFoT (FVD: 11.1) despite using the same VAE. The performance gain directly demonstrates the superiority of MiMo, as both methods share the same underlying video tokenization and differ primarily in their treatment of historical context. Qualitative examples are presented in Figure 4(a), where our method generates smooth, realistic continuations.

**Unconditional Video Generation (Uncond.)** For unconditional video generation on UCF-101, MiMo achieves remarkable results, establishing new state-of-the-art performance among AR approaches. Our method substantially outperforms the previous AR leader FAR by nearly 40 FVD points (240 vs 279) while utilizing the same DC-AE tokenizer, demonstrating the significant impact of our masked history modeling approach. Also noteworthy is the comparison with MAGI, which similarly employs Complete Teacher Forcing (CTF) during training—our method achieves a considerable performance improvement (FVD: 240 vs 298), validating the effectiveness of our masked history modeling objective. The generated videos exhibit diverse motions, realistic textures, and coherent temporal dynamics, as illustrated in the qualitative examples in Figure 4(b).

**Class-Conditional Video Generation (Cond.)** In class-conditional video generation on UCF-101, MiMo again demonstrates superior performance, achieving state-of-the-art results across AR

methods. Our approach surpasses FAR by 10 FVD points (98 vs 108), confirming the consistent benefits of our approach across different conditioning modalities. The comparison with ACDiT is also interesting—both methods utilize CTF and share similar architectural foundations, yet MiMo achieves notable improvements (FVD: 98 vs 111), consistent with our findings in unconditional generation when compared against MAGI. This consistency across tasks reinforces that our performance gains stem from improvements in history representation learning rather than task-specific optimizations. Representative generated videos are shown in Figure 4(c).

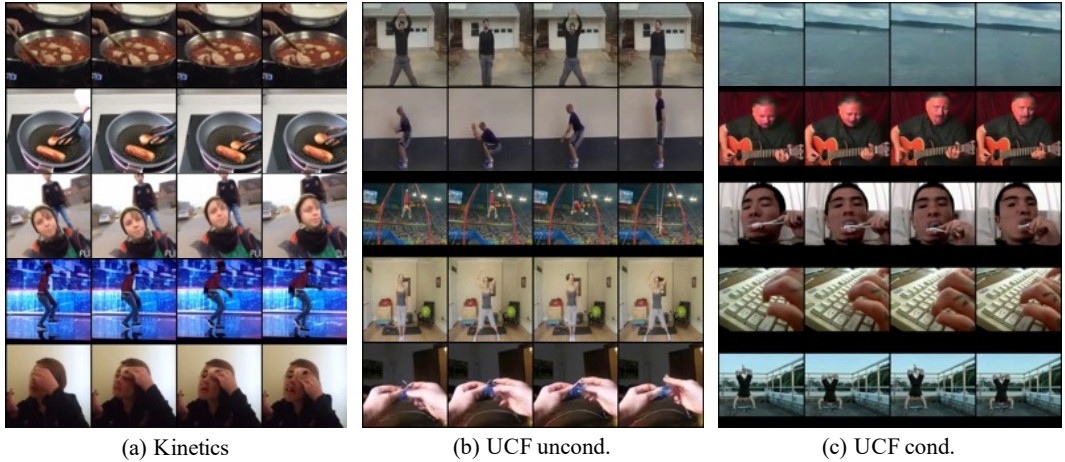

(a) Kinetics      (b) UCF uncond.      (c) UCF cond.

Figure 4: Visualization of generated videos.

## 4.3 ABLATION STUDY

This section ablates the designs of MiMo. All models are based on DiT-B trained on Kinetics for 100K steps with modifications and hyperparameters described in Section 4.1. ACDiT baseline is MiMo without masked history modeling, similar to MAGI and ACDiT.

Table 3: Comparison with variants of REPA.

| Method | | FVD↓ |
|---|---|---|
| ACDiT Baseline | | 54.814 |
| REPA | History | 40.022 |
| | Future | 40.253 |
| | Both | 36.542 |
| MiMo | | 36.601 |
| MiMo +REPA-Both | | 34.133 |

Table 4: Comparison of different prediction targets for masked history modeling.

| Target(s) | FVD↓ |
|---|---|
| ACDiT Baseline | 54.814 |
| Current Frame | 41.832 |
| Next Frame | 37.782 |
| Current + Next (MiMo) | 36.601 |
| Current + Next+NextNext | 36.263 |

**Comparison With REPA** An alternative way to inject good representations into the model is distilling the features from a VFM, known as REPA (Yu et al., 2024). Table 3 compares MiMo with several variants of REPA, supervising history frames, future frames, or both. Both REPA and MiMo can significantly improve convergence, while MiMo performs on par or better than all variants. In practice, however, VFMs for the user's domain of interest are not always available, in which cases MiMo is a viable substitute.

MiMo is complementary to VFM-based methods like REPA. MiMo excels at learning task-specific dynamics from the data, while VFM provides strong semantic priors. Combining MiMo with REPA in Table 3 yielded further improvements over either method alone. This suggests that MiMo and VFM capture different aspects of the data.

**Prediction Targets of Masked Modeling** One of the merits of MiMo is its flexibility: while diffusion always predicts the noise-free version of the noised current frame effectively, MiMo can

predict both the current and next frames for masked history modeling. Table 4 compares different prediction targets for masked modeling, and it is clear that predicting both current and history frames outperforms predicting either.

We also include a target of predicting the current and next two frames (Current + Next+NextNext) in Table 4. It is beneficial but yields diminishing returns. We hypothesize that predicting a more distant future frame is a significantly harder task, and the increased difficulty does not naively translate into proportional performance gains. Our proposed target (Current + Next) strikes an effective balance without the added complexity of longer-range prediction.

Table 5: Decoder position (placed after the $l$-th layer) moderately affects performance. DiT-B has 12 layers in total. None means CFT baseline.

| $l$ | None | 12 | 11 | 10 | 9 |
|---|---|---|---|---|---|
| FVD↓ | 54.814 | 36.601 | 35.815 | 35.838 | 37.593 |

Table 6: Architecture modifications.

| Modification | FVD↓ |
|---|---|
| Vanilla DiT | 37.763 |
| +RoPE | 37.313 |
| +Separate LayerNorm | 36.601 |

**Decoder Position**  The decoder for masked modeling is usually placed after the encoder (He et al., 2022a; Tong et al., 2022). We treat the first $l$ layers of the DiT model as the encoder, and shortcut the output of the $l$-the layer corresponding to the history frames into the decoder. Table 5 shows the effect of varying $l$. The performance is robust to $l$ when $l$ is close to the last layer.

**Model Architecture**  Table 6 shows the impact of architecture modifications on performance. Incorporating RoPE and separating LayerNorm layers for history frames both bring moderate gains.

Table 7: Ablations of hyperparameters $\lambda$ and mask ratios.

| $\lambda$ | 0.1 | 0.5 | 1.0 | 2.0 | Mask Ratios | [0.25, 0.25] | [0.25, 0.5] | [0.5, 0.75] |
|---|---|---|---|---|---|---|---|---|
| FVD↓ | 40.213 | 36.601 | 37.443 | 38.910 | FVD↓ | 37.539 | 36.601 | 39.121 |

**Hyperparamter Ablations**  Table 7 shows the impact of hyperparameters on MiMo-B models with different $\lambda$ from 0.25 to 2.0 and mask ratios from 0.25 to 0.75. For $\lambda$, a weight of 0.5 provides the best balance, but performance does not degrade sharply for nearby values. For mask ratios, performance remains relatively stable between 0.25 and 0.50; however, a higher mask ratio requires fine-tuning without masking to achieve better performance.

Table 8: Computational costs.

| Method | MiMo-XL | ACDiT-XL | FAR-XL |
|---|---|---|---|
| Wall-Clock Time | 0.750s | 0.788s | 0.704s |
| GFLOPs | 8.22 | 8.81 | 5.94 |

Table 9: Comparison of optimization strategies.

| Strategy | FVD↓ |
|---|---|
| ACDiT Baseline | 54.814 |
| Interleaving | 38.543 |
| MiMo | 36.601 |

**Computational Costs**  Table 8 shows the training computational costs of MiMo, ACDiT (our baseline), and FAR. Compared with our baseline (ACDiT), MiMo reduces training wall-clock time by 5%; compared with FAR, *MiMo increases training wall-clock time by a modest 10%, which is a small price for the significant performance boost* (25% from 279 to 240 on

Kinetics, and 14% from 279 to 240 on UCF-101). Compared with ACDiT, MiMo also reduces the FLOPs per training step due to masking (the decoders increase FLOPs). MiMo has higher training FLOPs compared with FAR, but the increase in training wall-clock time is moderate ( 10%) due to hardware acceleration. Note that MiMo has *no* additional inference cost once the training is complete.

**Alternative Optimization Strategy** While our approach of simply using a weighted sum of the two losses is standard practice for auxiliary loss training, an alternative approach is optimizing the diffusion loss and the auxiliary loss interleavingly (a diffusion-only training step is followed by a mask-only training step). The results are summarized in Table 9. While the interleaving approach has lower computational costs per step, it leads to slower convergence, which diminishes its speed gains.

## 5 RELATED WORKS

**Autoregressive Visual Generation.** Autoregressive language modeling (Radford, 2018; Radford et al., 2019) has facilitated the development of visual content generation using discrete visual tokens (Van Den Oord et al., 2017). In this framework, pre-trained visual tokenizers like VQ-VAE (Van Den Oord et al., 2017) map visual patches into a discrete latent space, allowing visual generation to be approached similarly to language modeling. Early works such as DALL-E (Ramesh et al., 2021) focus on text-to-image generation by learning a joint distribution between text and discrete image representations using an autoregressive cross-entropy loss. VideoGPT (Yan et al., 2021) extends this idea to video generation, employing discrete tokens for autoregressive video prediction. VideoPoet (Kondratyuk et al., 2023) further advances this approach by integrating a causal video tokenizer (Yu et al., 2023b). OmniTokenizer (Wang et al., 2024) proposes a unified tokenizer for both discrete and continuous representations. In contrast, our work focuses on frame-level causality rather than patch-level, avoiding the limitations of raster-scan order.

**Representations and Generative Modeling.** Recent advances in diffusion models highlight the importance of high-quality representations for generative modeling, as diffusion models inherently struggle to learn good representations (Yu et al., 2024; Zhang et al., 2025; Jiang et al., 2025; Wang & He, 2025). In practice, diffusion models are predominantly conditional generative models, where the conditions can be text prompts in T2I/T2v T2V generation, or history frames in VideoAR. Despite this prevalence, few studies have investigated how the quality of condition representations affects generative performance. Existing evidence from text-conditional generation provides compelling support for exploring this relationship. Replacing CLIP text encoders with large language models such as T5 and Llama has consistently improved generation quality, particularly for attributes strongly correlated with text conditions (counting, object reference, text rendering, etc.) (Esser et al., 2024; Gao et al., 2024; Kong et al., 2024; Hu et al., 2024b). Another evidence is that distilling class representations improves the performance of class-conditioned image generation (Wu et al., 2025). These observations naturally extend to VideoAR, where future frames depend on history frames as conditions, suggesting that enhanced history frame representations may potentially benefit video generation performance, which is the focus of our work.

## 6 CONCLUSION

In this work, we explored the fundamental question of whether good representations of history frames can improve VideoAR performance. Through systematic analysis, we demonstrated that enhancing history frame representations significantly benefits VideoAR, a finding that cannot be achieved by solely refining noisy future frames. Motivated by these insights, we proposed MiMo (Masked History Modeling), a novel framework that naturally integrates masked modeling into diffusion-based VideoAR. By applying masks to history frame tokens and training the model to predict masked tokens of current and future frames alongside denoising tasks, MiMo learns robust representations that improve VideoAR performance. Our approach requires no VFM or special architectural modifications. Extensive experiments across multiple benchmarks demonstrate that MiMo achieves competitive performance in video prediction and generation tasks, establishing new

state-of-the-art results. Notably, our framework substantially improves training efficiency and generation quality, showcasing the effectiveness of unified representation learning and diffusion modeling.

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

# A  DIFFUSION MODELING

In this section, we present a brief overview of diffusion-based generative models. These models learn to approximate target distributions through training denoising neural networks. There are two correlated approaches: "conventional" diffusion models based on score matching (Appendix A.1), and flow matching (Appendix A.2), introduced below.

## A.1  SCORE MATCHING

Diffusion models based on score matching (Ho et al., 2020; Kingma et al., 2021; Song et al., 2020b) generate samples $x \sim p_0(\cdot)$ by learning to invert a noise corruption process (i.e., the diffusion process) that transforms the data distribution into standard Gaussian noise $\epsilon \sim \mathcal{N}(0, I)$. The forward diffusion process is defined as:

$$p_\tau(x^{(\tau)}|x) = \mathcal{N}(\alpha_\tau x, \sigma_\tau^2 I); \quad \tau \in [0, 1], \tag{7}$$

where the coefficients $\alpha_\tau$ and $\sigma_\tau$ specify the "noise schedule" that interpolates between data and noise. Usually $\alpha_0 = 1, \sigma_0 = 0$ and $\alpha_1 = 0, \sigma_1 = 1$, so that $x^{(0)} = x$ and $x^{(1)} = \epsilon$.

The generative process is realized by integrating the reverse-time stochastic differential equation (SDE) (Song et al., 2020b; Lu et al., 2022) that describes the backward diffusion process:

$$dx^{(\tau)} = \left[ f(\tau) x^{(\tau)} - g^2(\tau) \nabla_{x^{(\tau)}} \log p_\tau(x^{(\tau)}) \right] d\tau + g(\tau) d\bar{w}_\tau, \tag{8}$$

where $\bar{w}_\tau$ denotes the reverse-time Wiener process, and the drift and diffusion coefficients $f$ and $g$ are given by:

$$f(\tau) = \frac{d \log \alpha_\tau}{d\tau}, \quad g^2(\tau) = -\sigma_\tau^2 \frac{d \log(\alpha_\tau/\sigma_\tau)}{d\tau}. \tag{9}$$

A score network $s_\theta(x^{(\tau)}; \tau)$ is trained to approximate $\nabla_{x^{(\tau)}} \log p_\tau(x^{(\tau)})$ via denoising score matching (Vincent, 2011):

$$\min_\theta \ \mathbb{E}_{\tau, \epsilon, x^{(0)}, x^{(\tau)}} \left[ \|\epsilon + \sigma_\tau s_\theta(x^{(\tau)}; \tau)\|_2^2 \right]. \tag{10}$$

Beyond directly modeling the score $s_\theta(x^{(\tau)}; \tau)$, diffusion models commonly use equivalent parameterizations tied to the forward relation $x^{(\tau)} = \alpha_\tau x^{(0)} + \sigma_\tau \epsilon$.

**Noise prediction (Ho et al., 2020).** The model predicts the noise $\epsilon_\theta(x^{(\tau)}; \tau) \approx \epsilon$, yielding the score via

$$s_\theta(x^{(\tau)}; \tau) = -\frac{1}{\sigma_\tau} \epsilon_\theta(x^{(\tau)}; \tau), \tag{11}$$

and is trained with the MSE objective $\mathbb{E} \left[ \|\epsilon - \epsilon_\theta(x^{(\tau)}; \tau)\|_2^2 \right]$.

**Data (clean sample) prediction (Sohl-Dickstein et al., 2015).** The model outputs a denoised estimate $x_\theta(x^{(\tau)}; \tau) \approx x^{(0)}$. Converting to a score gives

$$s_\theta(x^{(\tau)}; \tau) = -\frac{x^{(\tau)} - \alpha_\tau x_\theta(x^{(\tau)}; \tau)}{\sigma_\tau^2}, \tag{12}$$

which is equivalent to first forming $\hat{\epsilon} = (x^{(\tau)} - \alpha_\tau x_\theta)/\sigma_\tau$ and then using $s_\theta = -\hat{\epsilon}/\sigma_\tau$. Training objectives is minimizing $\mathbb{E} \left[ \|x^{(0)} - x_\theta(x^{(\tau)}; \tau)\|_2^2 \right]$.

**v-prediction (Salimans & Ho, 2022).** A time-dependent linear combination is predicted:

$$v_\theta(x^{(\tau)}; \tau) \approx \alpha_\tau \epsilon - \sigma_\tau x^{(0)}. \tag{13}$$

From $v_\theta$ one can recover all other targets:

$$\hat{\epsilon}(x^{(\tau)}; \tau) = \frac{\sigma_\tau x^{(\tau)} + \alpha_\tau v_\theta(x^{(\tau)}; \tau)}{\alpha_\tau^2 + \sigma_\tau^2}, \tag{14}$$

$$x_\theta(x^{(\tau)}; \tau) = \frac{\alpha_\tau x^{(\tau)} - \sigma_\tau v_\theta(x^{(\tau)}; \tau)}{\alpha_\tau^2 + \sigma_\tau^2}, \tag{15}$$

$$s_\theta(x^{(\tau)}; \tau) = -\frac{1}{\sigma_\tau} \hat{\epsilon}(x^{(\tau)}; \tau). \tag{16}$$

The training objective becomes $\mathbb{E}\left[\|\alpha_\tau \epsilon - \sigma_\tau x^{(0)} - v_\theta(x^{(\tau)}; \tau)\|_2^2\right]$.

All these parameterizations are connected by $\tau$-dependent linear transforms, and thus represent the same model class. Choosing among them mainly affects optimization stability and the weighting of errors across noise levels.

## A.2  FLOW MATCHING

Flow matching (Lipman et al., 2022; Liu et al., 2022; Esser et al., 2024) simplifies score matching by defining the generative process via ordinary differential equations (ODEs). Specifically, given the same definitions of $x, x^{(\tau)}, \alpha_\tau, \sigma_\tau$ as in Appendix A.1, the generative process is governed by a probability flow ODE:

$$\frac{\mathrm{d}x^{(\tau)}}{\mathrm{d}\tau} = v(x^{(\tau)}; \tau), \tag{17}$$

where the velocity field [3] $v(x^{(\tau)}; \tau)$ satisfies:

$$v(x^{(\tau)}, \tau) = \mathbb{E}\left[\frac{\mathrm{d}x^{(\tau)}}{\mathrm{d}\tau}\bigg|x^{(\tau)}\right] = \dot{\alpha}_\tau \mathbb{E}[x^{(0)}|x^{(\tau)}] + \dot{\sigma}_\tau \mathbb{E}[\epsilon|x^{(\tau)}], \tag{18}$$

where $\dot{\alpha}_\tau = \frac{\mathrm{d}\alpha_\tau}{\mathrm{d}\tau}$ and $\dot{\sigma}_\tau = \frac{\mathrm{d}\sigma_\tau}{\mathrm{d}\tau}$.

The flow matching objective trains a neural network $v_\theta(x^{(\tau)}; \tau)$ to minimize:

$$\min_\theta \mathbb{E}_{\tau, \epsilon, x^{(0)}, x^{(\tau)}}\left[\|v_\theta(x^{(\tau)}; \tau) - (\dot{\alpha}_\tau x^{(0)} + \dot{\sigma}_\tau \epsilon)\|_2^2\right]. \tag{19}$$

Flow matching and score matching are connected by the score function:

$$s(x^{(\tau)}; \tau) = -\frac{1}{\sigma_t}\mathbb{E}[\epsilon|x^{(\tau)}], \tag{20}$$

which corresponds to an equivalent reverse-time SDE (Ma et al., 2024a):

$$\mathrm{d}x^{(\tau)} = v(x^{(\tau)}; \tau)\mathrm{d}\tau - \frac{1}{2}\eta_\tau s(x^{(\tau)}; \tau)\mathrm{d}\tau + \sqrt{\eta_\tau}\mathrm{d}\bar{w}_t, \tag{21}$$

where $\eta_\tau$ controls the amount of stochasticity and $\bar{w}_t$ is a reverse-time Wiener process as in Appendix A.1. Solving Equations (18) and (20), we obtain:

$$s(x^{(\tau)}, \tau) = \frac{1}{\sigma_\tau} \cdot \frac{\alpha_\tau v(x^{(\tau)}, \tau) - \dot{\alpha}_\tau x^{(\tau)}}{\dot{\alpha}_\tau \sigma_\tau - \alpha_\tau \dot{\sigma}_\tau}. \tag{22}$$

Thus, flow matching and score matching learn the same model class.

Flow matching is easy to implement and usually converges faster than score matching in practice (Liu et al., 2022; Esser et al., 2024). Another advantage of flow matching is the flexibility to choose the diffusion coefficient $\eta_\tau$ independently of the training process, allowing for post-hoc optimization of the sampling procedure.

## B  EXTENDED RELATED WORKS

**Masked and Diffusion Video Generation**  Diffusion models have recently gained prominence in visual generation tasks (Ho et al., 2020; Rombach et al., 2022; He et al., 2022b; Guo et al., 2023; Chen et al., 2023; Guo et al., 2024), effectively extending to video generation. Video diffusion models (Brooks et al., 2024; Ho et al., 2022) utilize bidirectional attention and binary mask embeddings to facilitate frame-level autoregressive prediction. Notable works such as GameNGen (Valevski et al., 2024) use bidirectional diffusion models for real-time game generation. However, due to their bidirectional nature, these models cannot leverage KV Cache for extended video generation, limiting their scalability. Several masked video generators, such as Genie (Bruce et al., 2024), extend MaskGIT (Chang et al., 2022) into a causal-attention-based architecture for video generation. Despite their advantages, these methods suffer from the training-inference gap inherent in masked autoregressive modeling, which negatively impacts generation quality. In contrast, our approach fully leverages KV Cache during inference, facilitated by our training paradigm that bridges the training-inference gap through a novel complete teacher forcing paradigm.

---

[3]The velocity field in flow matching is *different* from the v-prediction parameterization in score matching, though they are correlated: the two parameterizations are connected by Equation (22).

Table 10: Hyperparameters.

| Name | MiMo-B | | MiMo-XL | |
|---|---|---|---|---|
| **Input** | | | | |
| Dataset | Kinetics-600 | Kinetics-600 | UCF-101 | UCF-101 |
| Task | prediction | prediction | class cond. | uncond. |
| Input shape | $17 \times 128 \times 128$ | $17 \times 128 \times 128$ | $16 \times 256 \times 256$ | $16 \times 256 \times 256$ |
| **VAE** | | | | |
| Compression ratio | $4 \times 8 \times 8$ | $4 \times 8 \times 8$ | $32 \times 32$ | $32 \times 32$ |
| Latent shape | $5 \times 16 \times 16$ | $5 \times 16 \times 16$ | $16 \times 8 \times 8$ | $16 \times 8 \times 8$ |
| **Architecture** | | | | |
| Patch size | $1 \times 1 \times 1$ | $1 \times 1 \times 1$ | $1 \times 1 \times 1$ | $1 \times 1 \times 1$ |
| Depth | 12 | 28 | 28 | 28 |
| Embed dim | 768 | 1152 | 1152 | 1152 |
| Num heads | 12 | 16 | 16 | 16 |
| RoPE theta | 100 | 100 | 100 | 100 |
| **Decoder** | | | | |
| Depth | 4 | 4 | 4 | 4 |
| $l$ | 9 | 23 | 23 | 23 |
| **Diffusion** | | | | |
| Parameterization | v-prediction | v-prediction | velocity | velocity |
| Noise scheduler | linear | linear | rectified flow | rectified flow |
| Weighting | fused min-SNR $\gamma = 5.0, \rho = 0.96$ | fused min-SNR $\gamma = 5.0, \rho = 0.96$ | logit-normal | logit-normal |
| Sampler | DDIM | DDIM | Euler | Euler |
| Sampling steps | 50 | 50 | 50 | 50 |
| Guidance | history guidance 1.05 | history guidance 1.05 | — | class guidance 2.0 |
| **Optimization** | | | | |
| Training steps | 100K | 360K | 180K | 180K |
| Batch size | 256 | 256 | 128 | 128 |
| Optimizer | AdamW | AdamW | AdamW | AdamW |
| Learning rate (LR) | $8 \times 10^{-4}$ | $8 \times 10^{-4}$ | $4 \times 10^{-4}$ | $4 \times 10^{-4}$ |
| Warmup steps | 10K | 10K | 10K | 10K |
| LR schedule | cosine | cosine | cosine | cosine |
| End LR | $10^{-5}$ | $10^{-5}$ | $10^{-5}$ | $10^{-5}$ |
| Weight decay | 0.001 | 0.001 | 0.001 | 0.001 |
| $(\beta_1, \beta_2)$ | (0.9, 0.99) | (0.9, 0.99) | (0.9, 0.99) | (0.9, 0.99) |
| Gradient clipping | 1.0 | 1.0 | 1.0 | 1.0 |
| $\lambda$ | 0.5 | 0.5 | 0.5 | 0.5 |
| Mask ratios | $[0.25, 0.5]$ | $[0.25, 0.5]$ | $[0.25, 0.5]$ | $[0.25, 0.5]$ |
| EMA decay | 0.999 | 0.999 | 0.9999 | 0.9999 |

## C  DATASETS

**Kinetics-600**  Kinetics-600 (Carreira et al., 2018) is a large-scale video action recognition dataset that extends the original Kinetics-400 dataset (Kay et al., 2017), containing approximately 500,000 video clips across 600 human action categories, sourced from YouTube and covering diverse human actions ranging from sports and cooking to dancing and musical performances. The dataset is split into training, validation, and test sets, with each action class containing at least 600 video clips in the training set and 50 clips in both validation and test sets. Videos in Kinetics-600 are characterized by

their temporal dynamics and complex motion patterns, making it a challenging benchmark for video understanding tasks. The dataset provides rich temporal information and diverse visual content, which makes it particularly suitable for evaluating autoregressive video modeling approaches that need to capture long-term temporal dependencies and generate coherent future frames based on historical context. Following existing works (Song et al., 2025), we use a resolution of $128 \times 128$ pixels and train on the training set while evaluating on the test set. The model is conditioned on the first 5 frames and predicts the next 11 frames, totaling 16 frames.

**UCF-101**  UCF-101 (Soomro et al., 2012) is a widely used action recognition dataset consisting of 13,320 video clips distributed across 101 action categories. The dataset was collected from YouTube and contains realistic videos with significant variations in camera motion, object appearance, pose, scale, viewpoint, cluttered background, and illumination conditions. Each action class contains 25 groups of videos, with each group sharing common features such as similar backgrounds, similar viewpoints, etc. UCF-101 covers a diverse range of human actions, including sports activities (e.g., basketball, tennis, surfing), musical instrument playing, and daily life activities. Despite being smaller in scale compared to Kinetics datasets, UCF-101 remains a fundamental benchmark for assessing the generalization capability of video models across different domains and action complexities, due to its well-curated action categories. We follow the protocol of Gu et al. (2025) and use a resolution of $256 \times 256$ pixels. The models are trained on the full UCF-101 dataset and evaluated with class labels as the only initial condition, generating a total of 16 frames.

## D  Implementation Details

Table 10 summarizes the hyperparameters we use in our implementations. The details are discussed in the following sections.

### D.1  Model Architectures

**Diffusion Models**  We employ the Diffusion Transformer (DiT) (Peebles & Xie, 2023) with full 3D attention as our backbone. The DiT block is analogous to a vision transformer (ViT) (Dosovitskiy et al., 2020) block and replaces the LayerNorm (Ba et al., 2016) layers with adaptive Layer-Norm (AdaLN) (Peebles & Xie, 2023) layers to inject diffusion timestep condition into the features. AdaLN works by embedding the timesteps using sinusoidal positional encoding (Vaswani et al., 2017) and feeding them to an MLP to predict the shift and bias factors for LayerNorm layers. For class-conditioned generation, the class labels are also embedded and added into the timestep embeddings as additional conditions. Following existing works (Song et al., 2025; Gu et al., 2025; Hu et al., 2024a), AdaLN is applied separately to each noisy future frame because different frames can have different diffusion timesteps during training (Section 3.3). We use QK normalization (Henry et al., 2020) to stabilize training. Below, we introduce two other modifications we apply to vanilla DiT: separate LayerNorm and 3D RoPE. The vanilla DiT block and our modified DiT block are illustrated in Figure 5.

Separate LayerNorm layers instead of AdaLN are applied to the clean frames. Note that the other parameters are shared among all frames regardless of whether they are history or future frames. The attention mask introduced in Section 3.3 is applied to the attention operation to ensure causality between history and future frames.

Additionally, we incorporate axial 3D RoPE (Su et al., 2024) and assign an equal number of channels to encode the positions along the $T, H, W$ dimensions.

**VAE and Patch Size of DiT**  Video generation models usually work in some compressed latent space with reduced space-time dimensions to save computations, due to the sheer volume of pixels in video. In all of our experiments, the patch size of DiT is $1 \times 1 \times 1$ $(T, H, W)$, meaning that compression is done solely in VAE.

For fair comparison with existing methods, we adopt the pretrained 3D video VAE of DFoT (Song et al., 2025) for Kinetics-600 experiments. DFoT's VAE has a compression ratio of $4 \times 8 \times 8$ $(T, H, W)$, where the first frames are separately encoded and the following frames are temporally

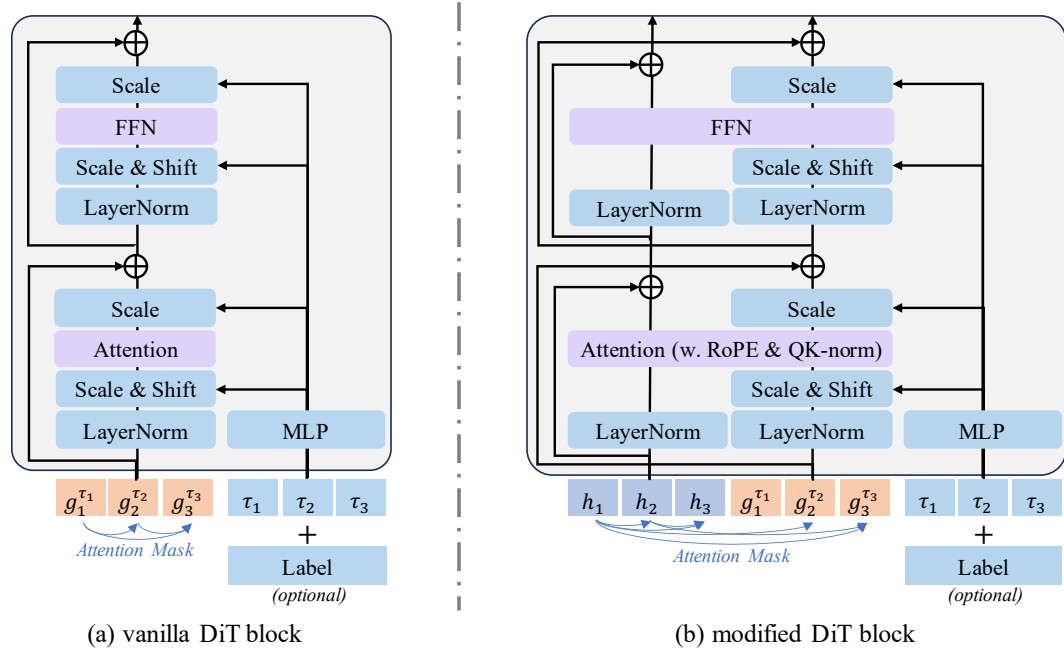

(a) vanilla DiT block         (b) modified DiT block

Figure 5: Illustration of vanilla and our modified DiT blocks.

downsampled by a factor of 4, following Yu et al. (2023a). The input resolution is $128 \times 128$ pixels with 17 frames, leading to a latent shape of $5 \times 16 \times 16$ per video clip.

We utilize the 2D image DC-AE of FAR (Gu et al., 2025) for UCF-101 experiments. FAR's DC-AE has a compression ratio of $32 \times 32$ with no temporal compression, and it encodes each frame independently. Given input of 16 frames with a resolution of $256 \times 256$, the latent shape is $16 \times 8 \times 8$.

**Decoder** The decoders take masked history frame features from intermediate DiT layers as the only input, and fill the masked positions with learnable query tokens. Then, the input is fed into a stack of several decoder blocks and reshaped to the same dimensions as the latents of the history (or future) frames as the output. The decoder block is the vanilla ViT block with axial 3D RoPE.

### D.2 DIFFUSION

**Kinetics-600 Experiments** For Kinetics-600, we use a linear noise schedule (Nichol & Dhariwal, 2021) with the v-prediction parameterization (Salimans & Ho, 2022) and zero terminal SNR (Lin et al., 2024). We use the DDIM sampler (Song et al., 2020a) with 50 sampling steps during inference. We also incorporate the fused min-SNR loss weighting (Chen et al., 2024a), a variant of min-SNR loss weighting (Hang et al., 2023) for video diffusion, to accelerate convergence.

Fused min-SNR extends the standard min-SNR loss weighting by accounting for the "signals" from previous frames. The difference between the two methods is the way to compute the signal-to-noise ratio (SNR) used for loss weighting. Using the notations in Section 2, SNR is defined as $\text{SNR}_\tau = \alpha_\tau^2 / \sigma_\tau^2$. Fused min-SNR first normalizes the SNR to $[0, 1]$ by dividing by the maximal value of SNR. Since min-SNR weighting clips the SNR value with the hyperparameter $\gamma > 0$, we normalize by $\gamma$. Then, it computes fused SNR $S_t'$ with decaying factor $\rho > 0$:

$$S_t = \text{normalized SNR factor for the } t\text{-th noisy future frame} \in [0, 1], \tag{23}$$

$$\bar{S}_t = \rho \bar{S}_{t-1} + (1 - \rho) S_t \quad \text{(exponentially decayed cumulative SNR)}, \tag{24}$$

$$S_t' = 1 - (1 - S_t)(1 - \bar{S}_{t-1}) \quad \text{(fused reweighting factor)}, \tag{25}$$

Fused SNR $S_t'$ combines the current frame signal with accumulated history signals, treating them as independent probabilistic events. This accounts for the additional information available from history

```python
def compute_loss_weight(snr, gamma, prediction_type, decay=None,
    causal=True):
    """Compute SNR weighting.

    Args:
        snr (torch.Tensor): per-frame SNR of shape [B, T]
        gamma (float): clip threshold of min-SNR
        prediction_type (str): "epsilon", "v_prediction", or "sample"
        decay (float, optional): if not None, enable fused min-SNR with
            the specified decay factor
        causal (bool, optional): whether we are training a causal model

    Returns:
        weight (torch.Tensor): per-frame loss weight of shape [B, T]
    """
    # Compute fused SNR
    clipped_snr = snr.clamp(max=gamma)
    if decay is not None:
        normalized_clipped_snr = clipped_snr / gamma
        normalized_snr = snr / gamma

        def compute_cum_snr(reverse: bool = False):
            new_normalized_clipped_snr = (
                normalized_clipped_snr.flip(1)
                if reverse
                else normalized_clipped_snr
            )
            cum_snr = torch.zeros_like(new_normalized_clipped_snr)
            for t in range(0, snr.shape[1]):
                if t == 0:
                    cum_snr[:, t] = new_normalized_clipped_snr[:, t]
                else:
                    cum_snr[:, t] = (
                        decay * cum_snr[:, t - 1]
                        + (1 - decay) * new_normalized_clipped_snr[:, t]
                    )
            cum_snr = torch.nn.functional.pad(cum_snr[:, :-1], (1, 0, 0,
                0), value=0.0)
            return cum_snr.flip(1) if reverse else cum_snr

        if causal:
            cum_snr = compute_cum_snr()
        else:
            cum_snr = compute_cum_snr(reverse=True) + compute_cum_snr()
            cum_snr *= 0.5
        clipped_fused_snr = 1 - (1 - cum_snr * decay) * (1 -
            normalized_clipped_snr)
        fused_snr = 1 - (1 - cum_snr * decay) * (1 - normalized_snr)
        clipped_snr = clipped_fused_snr * gamma
        snr = fused_snr * gamma

    # Compute loss weight
    if prediction_type == "epsilon":  # noise-prediction
        weight = clipped_snr / snr
    elif prediction_type == "v_prediction":  # v-prediction
        weight = clipped_snr / (snr + 1)
    else:  # data-prediction
        weight = clipped_snr

    return weight
```

Listing 1: Fused min-SNR (PyTorch psuedo-code)

context in video generation, beyond what standard SNR weighting captures. The $S'_t$ is denormalized by multiplying $\gamma$ and used to compute the loss weighting as normal min-SNR weighting does.

Listing 1 summarizes the algorithm to compute fused min-SNR weighting.

**UCF-101 Experiments** For UCF-101, we follow Gu et al. (2025) and use flow matching (Liu et al., 2022; Lipman et al., 2022; Albergo & Vanden-Eijnden, 2022) with a "straigh" flow path, i.e., $\alpha_\tau = 1 - \tau, \sigma_\tau = \tau$. We also adopt the logit-normal timestep sampling strategy (Esser et al., 2024), where the timesteps $\tau$ are sampled from a logit-normal distribution (instead of uniformly):

$$\pi_{\ln}(\tau) = \frac{1}{\sqrt{2\pi}} \frac{1}{t(1-t)} \exp\left(-\frac{\log^2(t/1-t)}{2}\right). \tag{26}$$

We use the Euler integrator sampler (Esser et al., 2024) with 50 sampling steps during inference.

## D.3 TRAINING

---

**Algorithm 1** Training (v-prediction or flow matching)

---

**Input:** Dataset $\mathcal{D}$, noise schedule $\{(\alpha_\tau, \sigma_\tau)\}_\tau$, velocity network $v_\theta$, decoder $\varphi_\theta$, loss weight $\lambda$
**Output:** Trained velocity network $v_\theta$
1: **while** not converged **do**
2:     Sample video clip $\mathbf{x} = \{x_t\}_{t=1}^T$ from $\mathcal{D}$
3:     $\mathbf{h} \leftarrow \mathbf{x}, \mathbf{f} \leftarrow \mathbf{x}$                                    // Duplicate $\mathbf{x}$ as history $\mathbf{h}$ and future $\mathbf{f}$
4:     Sample $\{\tau_t \sim \text{Uniform}[0,1]\}_{t=1}^T$ and $\{\epsilon_t \sim \mathcal{N}(0, I)\}_{t=1}^T$
5:     Sample random tube mask $\mathcal{M}$
6:     $h_t^\mathcal{M} \leftarrow \text{applyMask}(h_t, \mathcal{M})$                          // Apply $\mathcal{M}$ to history frame
7:     $\mathcal{L} \leftarrow 0$
8:     **for** $t = 1$ to $T$ **do**
9:         $f_t^{(\tau_t)} \leftarrow \alpha_{\tau_t} f_t + \sigma_{\tau_t} \epsilon_t$                  // Add noise to future frame
10:        $v_{\text{target}} \leftarrow \alpha_{\tau_t} \epsilon_t - \sigma_{\tau_t} f_t$ or $v_{\text{target}} \leftarrow \dot{\alpha}_{\tau_t} f_t + \dot{\sigma}_{\tau_t} \epsilon_t$    // v-prediction or flow matching
11:        $v_{\text{pred}} \leftarrow v_\theta(f_t^{(\tau)}; \tau, h_{1:t}^\mathcal{M})$         // Internally apply attention mask (Figure 3(a))
12:        $\mathcal{L}_{\text{diff}} \leftarrow \|v_{\text{pred}} - v_{\text{target}}\|_2^2$             // Diffusion loss (Equation (4))
13:        $v_{\text{feat}}^{h,l} \leftarrow v_\theta^{h,l}(f_t^{(\tau)}; \tau, h_{1:t}^\mathcal{M})$      // Output features of the $l$-th layer for history frames $h_{1:t}$
14:        $\mathcal{T}_t \leftarrow \{t, t+1\}$                                   // Frame indexes
15:        $\mathcal{L}_{\text{mask}} \leftarrow \frac{1}{|\mathcal{T}_t|} \sum_{t' \in \mathcal{T}_t} \|h_{t'} - \varphi_\theta(t' - t, v_{\text{feat}}^{h,l})\|_2^2$    // Masked history modeling loss (Equation (5))
16:        $\mathcal{L} \leftarrow \mathcal{L} + \mathcal{L}_{\text{diff}} + \lambda \mathcal{L}_{\text{mask}}$
17:     **end for**
18:     Update $\theta$ using $\nabla_\theta \mathcal{L}$
19: **end while**

---

The training algorithm is summarized in Algorithm 1. Training hyperparameters are summarized in Table 10. Note that the masked modeling loss (Equation (5)) is computed in the latent space between the latents of the history (or future) frames and the predictions of the decoders.

## D.4 INFERENCE

The sampling algorithm is summarized in Algorithm 2. Inference hyperparameters are summarized in Table 10. The inference process is the same as in other diffusion-based video generation models (Song et al., 2025; Gu et al., 2025; Hu et al., 2024a): Given initial conditions (initial frames for video prediction, class labels for class-conditioned generation, or no condition for unconditioned generation), the model iteratively denoises the next frame starting from pure noise, and appends the generated frame after the known (provided as initial conditions or generated) frames until all frames are known.

---

**Algorithm 2** Sampling (v-prediction or flow matching)

---

**Input:** Noise schedule $\{(\alpha_\tau, \sigma_\tau)\}_\tau$, sampling steps $N$, velocity network $v_\theta$, initial frames $x_{1:t_0}$ ($\varnothing$ if $t_0 = 0$)
**Output:** Clean frames $x_{1:T}$
1: **for** $t = t_0 + 1$ to $T$ **do**
2:     $x_t \sim \mathcal{N}(0, I)$                // Initialize with noise at $\tau = \tau_N = 1$
3:     **for** $i = N$ to $1$ **do**
4:        $v_{\text{pred}} \leftarrow v_\theta(x_t; \tau_i, x_{1:t-1})$        // Internally apply block-causal attention mask
5:        $x_t \leftarrow \text{Sampler}(x_t; \tau_i, \tau_{i-1}, v_{\text{pred}})$        // Sampler step, $\tau_0 = 0$
6:     **end for**
7:     $x_{1:t} \leftarrow x_{1:t-1} + \{x_t\}$        // Append generated frame after known frames
8: **end for**

---

### D.5 HISTORY GUIDANCE

We incorporate a simplified version of history guidance (Song et al., 2025) into diffusion-based VideoAR. History guidance takes advantage of the insight that history frames are the conditions for generating the future frames, much like class labels as conditions for class-conditioned generation, and applies classifier-free guidance (CFG) (Ho & Salimans, 2022) with history frames as conditions. Adopting the notations in Section 2, history guidance modifies the score function as

$$s_\theta^w(x_{t+1}^{(\tau)}; \tau, x_{1:t}) = (1 - w) \cdot s_\theta(x_{t+1}^{(\tau)}; \tau, \varnothing) + w \cdot s_\theta(x_{t+1}^{(\tau)}; \tau, x_{1:t}), \tag{27}$$

where $\varnothing$ means no history frame and $w > 1$ is the guidance scale. We compute $s_\theta(x_{t+1}^{(\tau)}; \tau, \varnothing)$ by forbidding $(x_{t+1}^{(\tau)}$ to attend to $x_{1:t}$ via attention masking, i.e., by setting the corresponding noisy-to-clean rows in the attention mask (Figure 3(a)) to $-\infty$.

During training, we randomly select $r = 10\%$ future frames and forbid them from attending to the history frames. The training algorithm with history guidance is summarized in Algorithm 3.

During inference, $s_\theta^w(x_{t+1}^{(\tau)}; \tau, \varnothing)$ is computed by Equation (27) and the other process is the same as in normal CFG. The sampling algorithm with history guidance is summarized in Algorithm 4.

## E BASELINES

In our work, we primarily consider three baseline methods in Figure 1 and Section 3.2. All the considered baselines are trained with the *same* model architecture and hyperparameters as shown in Table 10 unless otherwise specified, with the only difference being the training strategies.

**ACDiT** ACDiT (Hu et al., 2024a) also adopts complete teacher forcing as in MiMo. The primary difference between MiMo and ACDiT is that we apply the masked history modeling target on the history frames. Thus, the direct comparison between ACDiT and MiMo clearly demonstrates the advantage of our approach and the benefit of good history representations.

**FAR** FAR (Gu et al., 2025) adopts diffusion forcing (Chen et al., 2024a; Song et al., 2025), it randomly replaces some noisy frames with their clean version to simulate clean history frames. The better performance of MiMo over FAR demonstrates that MiMo can achieve competitive performance even against the best performing models in a broader context.

**REPA** REPA (Yu et al., 2024) was originally proposed to improve the representation quality of the noisy images being denoised. We adapt it to diffusion-based VideoAR following the approach of Zhang et al. (2025). Compared with REPA, we focus on the representations of the history frames that serve as conditions in AR modeling, while REPA does not consider the AR context. Also, REPA requires a VFM, but sVFM may not always be available and may misbehave for out-of-distribution (OOD) data, while MiMo does not rely on VFM.

For the analysis in Section 3.2, we align the features of the 4-the layer with a pretrained VideoMAE-L (Tong et al., 2022) using a loss weight of $0.5$. The feature dimensions of DiT models and Video-

---

**Algorithm 3** Training with history guidance (v-prediction or flow matching)

---

**Input:** Dataset $\mathcal{D}$, noise schedule $\{(\alpha_\tau, \sigma_\tau)\}_\tau$, velocity network $v_\theta$, decoder $\varphi_\theta$, loss weight $\lambda$, drop rate $r$

**Output:** Trained velocity network $v_\theta$

1: **while** not converged **do**
2:      Sample video clip $\mathbf{x} = \{x_t\}_{t=1}^T$ from $\mathcal{D}$
3:      $\mathbf{h} \leftarrow \mathbf{x}, \mathbf{f} \leftarrow \mathbf{x}$           // Duplicate $\mathbf{x}$ as history $\mathbf{h}$ and future $\mathbf{f}$
4:      Sample $\{\tau_t \sim \text{Uniform}[0,1]\}_{t=1}^T$ and $\{\epsilon_t \sim \mathcal{N}(0, I)\}_{t=1}^T$
5:      Sample random tube mask $\mathcal{M}$
6:      $h_t^{\mathcal{M}} \leftarrow \text{applyMask}(h_t, \mathcal{M})$          // Apply $\mathcal{M}$ to history frame
7:      $\mathcal{L} \leftarrow 0$
8:      **for** $t = 1$ to $T$ **do**
9:          $f_t^{(\tau_t)} \leftarrow \alpha_{\tau_t} f_t + \sigma_{\tau_t} \epsilon_t$         // Add noise to future frame
10:        $v_{\text{target}} \leftarrow \alpha_{\tau_t} \epsilon_t - \sigma_{\tau_t} f_t$ or $v_{\text{target}} \leftarrow \dot{\alpha}_{\tau_t} f_t + \dot{\sigma}_{\tau_t} \epsilon_t$    // v-prediction or flow matching
11:          **if** $\text{Uniform}[0,1] < r$ **then**
12:             $v_{\text{pred}} \leftarrow v_\theta(f_t^{(\tau)}; \tau, \varnothing)$        // Randomly drop history frames
13:          **else**
14:             $v_{\text{pred}} \leftarrow v_\theta(f_t^{(\tau)}; \tau, h_{1:t}^{\mathcal{M}})$     // Internally apply attention mask (Figure 3(a))
15:          **end if**
16:          $\mathcal{L}_{\text{diff}} \leftarrow \|v_{\text{pred}} - v_{\text{target}}\|_2^2$        // Diffusion loss (Equation (4))
17:          $v_{\text{feat}}^{h,l} \leftarrow v_\theta^{h,l}(f_t^{(\tau)}; \tau, h_{1:t}^{\mathcal{M}})$    // Output features of the $l$-th layer for history frames $h_{1:t}$
18:          $\mathcal{T}_t \leftarrow \{t, t+1\}$            // Frame indexes
19:          $\mathcal{L}_{\text{mask}} \leftarrow \frac{1}{|\mathcal{T}_t|} \sum_{t' \in \mathcal{T}_t} \|h_{t'} - \varphi_\theta(t' - t, v_{\text{feat}}^{h,l})\|_2^2$    // Masked history modeling loss
         (Equation (5))
20:          $\mathcal{L} \leftarrow \mathcal{L} + \mathcal{L}_{\text{diff}} + \lambda \mathcal{L}_{\text{mask}}$
21:      **end for**
22:      Update $\theta$ using $\nabla_\theta \mathcal{L}$
23: **end while**

---

**Algorithm 4** Sampling with history guidance (v-prediction or flow matching)

---

**Input:** Noise schedule $\{(\alpha_\tau, \sigma_\tau)\}_\tau$, sampling steps $N$, guidance scale $w$, velocity network $v_\theta$, initial frames $x_{1:t_0}$ ($\varnothing$ if $t_0 = 0$)

**Output:** Clean frames $x_{1:T}$

1: **for** $t = t_0 + 1$ to $T$ **do**
2:     $x_t \sim \mathcal{N}(0, I)$           // Initialize with noise at $\tau = \tau_N = 1$
3:     **for** $i = N$ to $1$ **do**
4:        $v_{\text{pred}} \leftarrow v_\theta(x_t; \tau_i, x_{1:t-1})$      // Internally apply block-causal attention mask
5:        $v_\varnothing \leftarrow v_\theta(x_t; \tau_i, \varnothing)$          // Negative condition
6:        $v_{\text{pred}} \leftarrow (1 - w) \cdot v_\varnothing + w \cdot v_{\text{pred}}$      // Apply guidance
7:        $x_t \leftarrow \text{Sampler}(x_t; \tau_i, \tau_{i-1}, v_{\text{pred}})$      // Sampler step, $\tau_0 = 0$
8:     **end for**
9:     $x_{1:t} \leftarrow x_{1:t-1} + \{x_t\}$        // Append generated frame after known frames
10: **end for**

---

MAE are aligned following the strategy of Zhang et al. (2025), which interpolates the DiT's representations to match the feature dimensions of the pre-trained VideoMAE.

## F   EVLAUTION DETAILS

**Fréchet Video Distance (FVD)**   FVD (Unterthiner et al., 2018) is a perceptual metric designed to evaluate the quality of generated videos by measuring the distributional distance between real and generated video sequences. Similar to the Fréchet Inception Distance (FID) (Heusel et al., 2017) used for images, FVD employs a pre-trained 3D convolutional neural network (specifically, an Inflated 3D ConvNet or I3D model trained on Kinetics-400) (Carreira & Zisserman, 2017) to extract spatio-temporal features from video clips. Then Fréchet distance (Dowson & Landau, 1982)

is computed between the feature distributions of real and generated videos by fitting multivariate Gaussian distributions to the extracted features and calculating the Wasserstein-2 distance (Villani et al., 2008) between them. Lower FVD scores indicate higher similarity to real video distributions. FVD captures both spatial and temporal aspects of video content, making it a standard evaluation tool in video synthesis research. Following prior works (Song et al., 2025), we compute FVD for the entire video, including both initial conditioning frames (for Kinetics-600) and generated frames, to assess the overall consistency.

**Centered Kernel Nearest-Neighbor Alignment (CKNNA)**    CKNNA (Huh et al., 2024) is a non-parametric evaluation metric that measures the alignment between two sets of features by analyzing their local neighborhood structures. CKNNA relaxes the overly rigid Centered Kernel Alignment (CKA) (Kornblith et al., 2019) metric by computing similarity only for the mutual nearest neighbours of each feature vector. Given two sets of vectorized features $\{\phi_i \in \mathbb{R}^n\}$ and $\{\psi_i \in \mathbb{R}^m\}$ from two models and inner product operator $\langle \cdot, \cdot \rangle$, CKNNA first computes centered kernel matrices:

$$\bar{\mathbf{K}}_{ij} = \langle \phi_i, \phi_j \rangle - \mathbb{E}_l[\langle \phi_i, \phi_l \rangle], \quad \bar{\mathbf{L}}_{ij} = \langle \psi_i, \psi_j \rangle - \mathbb{E}_l[\langle \psi_i, \psi_l \rangle] \tag{28}$$

The centering operation removes the mean similarity, focusing on relative relationships rather than absolute magnitudes. CKNNA restricts the alignment computation to mutual nearest neighbors:

$$\mathsf{Align}_{\mathsf{knn}}(\mathbf{K}, \mathbf{L}) = \sum_i \sum_j \alpha(i,j) \cdot \bar{\mathbf{K}}_{ij} \bar{\mathbf{L}}_{ij} \tag{29}$$

$$\text{where} \quad \alpha(i,j) = \mathbf{1}[\phi_j \in \mathsf{knn}(\phi_i) \wedge \psi_j \in \mathsf{knn}(\psi_i) \wedge i \neq j] \tag{30}$$

The indicator function $\alpha(i,j)$ ensures we only consider sample pairs whose members are nearest neighbors to each other, emphasizing local structural consistency over global alignment. The final CKNNA metric is the normalized version:

$$\mathsf{CKNNA}(\mathbf{K}, \mathbf{L}) = \frac{\mathsf{Align}_{\mathsf{knn}}(\mathbf{K}, \mathbf{L})}{\sqrt{\mathsf{Align}_{\mathsf{knn}}(\mathbf{K}, \mathbf{K}) \cdot \mathsf{Align}_{\mathsf{knn}}(\mathbf{L}, \mathbf{L})}} \tag{31}$$

This normalization bounds the metric to $[0, 1]$, where higher values indicate better preservation of local neighborhood structure between the two representation spaces. Intuitively, CKNNA measures whether two feature representations maintain similar local similarity structures within their respective neighborhoods. Following prior works (Huh et al., 2024; Yu et al., 2024), we evaluate representation alignment using CKNNA with $k = 10$ nearest neighbors. We randomly sample 10,000 videos from the Kinetics-600 test set and extract globally average pooled features using both a pretrained VideoMAE-L (Tong et al., 2022) (as reference) and our models, treating all frames as clean history frames. Then, we compute CKNNA between the features of VideoMAE-L and features from each layer of the query models, reporting the highest alignment score across all layers.

**Linear Probing**    We follow the linear probing protocol of MAE (He et al., 2022a). Specifically, we use the model representations of the clean history frames for linear probing training and evaluation. Global average pooling is applied to the output feature map to obtain a single feature vector for each video. The feature vector is then fed to a parameter-free BatchNorm (Ioffe & Szegedy, 2015) layer and a linear classifier layer. The training batch size is 128, the learning rate is $10^{-3}$ and decayed to 0 with a cosine schedule, the weight decay is 0.01, and the training length is 10 epochs. Random flipping is used during training. Top-1 accuracy is reported.

## G    ADDITIONAL VISUALIZATIONS

### G.1    SAMPLES

This section shows the samples generated by MiMo on Kinetics-600 (Figure 6), UCF-101 class-conditioned generation (Figure 7), and unconditional generation (Figure 8). Each row is a generated video containing 16 frames.

### G.2    ATTENTION HEATMAPS

In Figures 9 and 10 we show the attention heatmaps (marked by red) of two videos, each without and with MiMo. The center position of the last frame (marked by a blue dot) serves as the query,

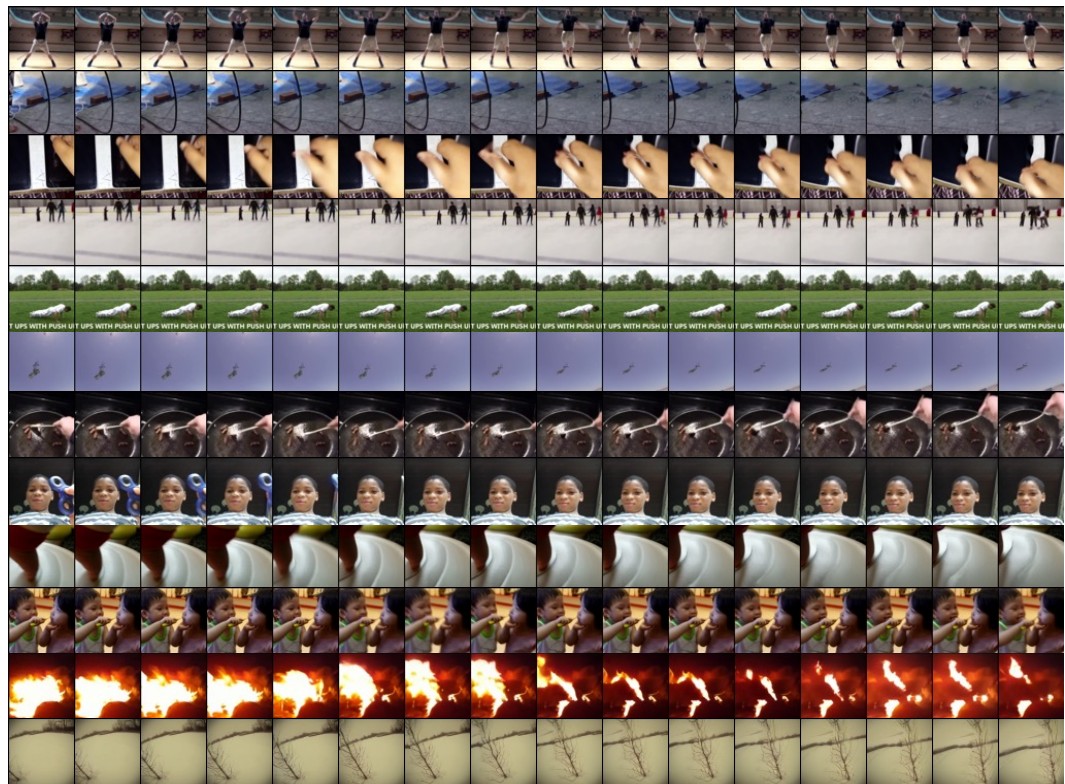

Figure 6: Uncurated samples of Kinetics history frames conditional generation.

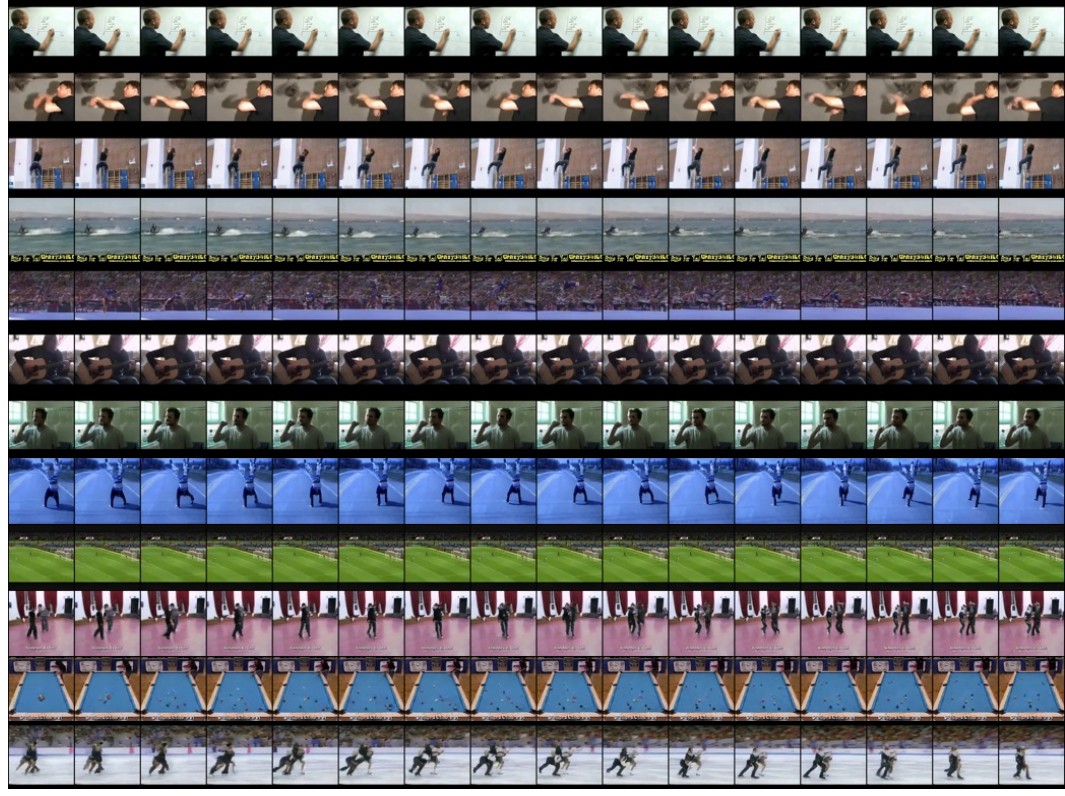

Figure 7: Uncurated samples of UCF-101 class-conditioned generation.

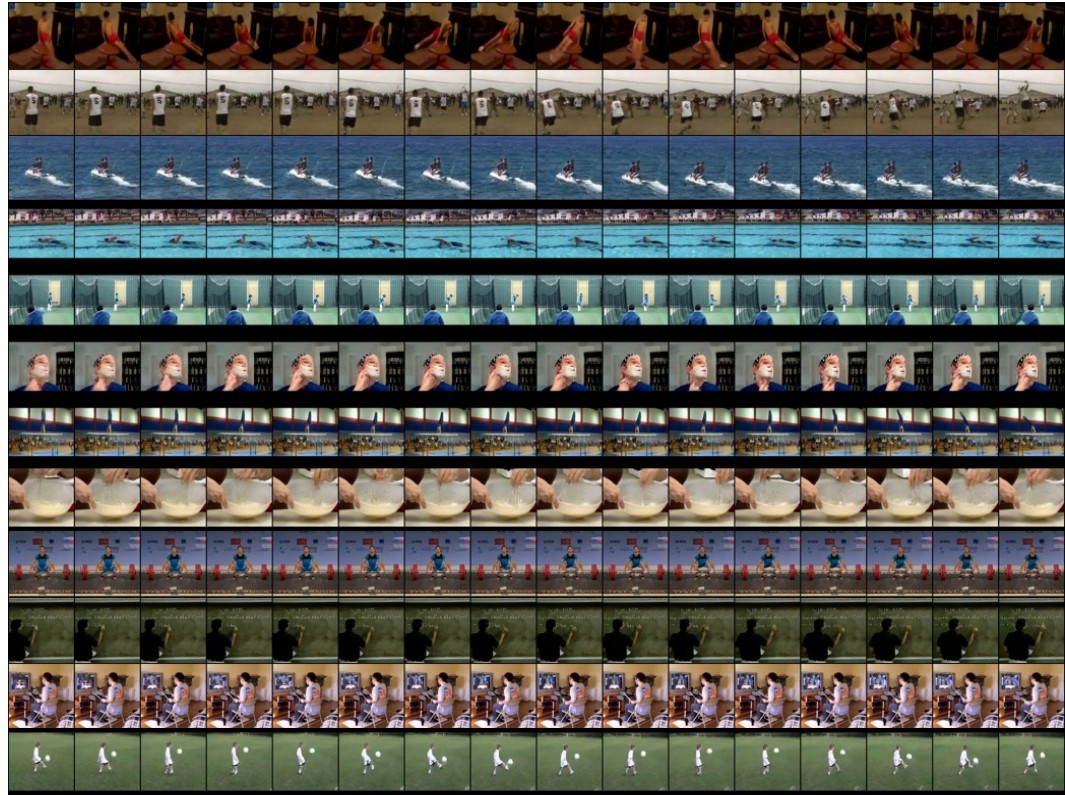

Figure 8: Uncurated samples of UCF-101 unconditional generation.

while the other positions are the keys. We take the attention weights from four random heads and layers 4, 8, 12, 16, 20.

As shown in Figures 9 and 10, without MiMo, attention patterns are more dispersed and less focused, whereas with MiMo, attention heatmaps show more concentrated patterns that exhibit stronger semantic correlations with the query content. Additionally, Figure 10 demonstrates how different transformer layers specialize in matching distinct body parts (e.g., arms, torso, legs), revealing the hierarchical nature of the learned representations.

### G.3 Embedding Visualization

Figure 11 shows the UMAP (McInnes et al., 2018) visualization of video embeddings without and with MiMo. The model is a DiT-XL and is trained for 360K steps on Kinetics-600. As shown in Figure 11, without MiMo, the distribution of video embeddings is mostly uniform, while MiMo introduces some structures in the embedding distribution by learning a more structured representation space.

## H  Additional Experiments

### H.1 Complementary Evaluation Metrics

Our evaluation mainly relies on the standard FVD metric. However, FVD is known to have several issues and may not fully capture real-world video dynamics (Luo et al., 2024). To provide a more comprehensive evaluation of MiMo's performance, we adopt two additional metrics as complements.

**VMMD**  Our VMMD (V-JEPA 2 Maximum Mean Discrepancy) metric is based on the CMMD metric (Jayasumana et al., 2024). The VMMD metric benchmarks the perceptual similarity between

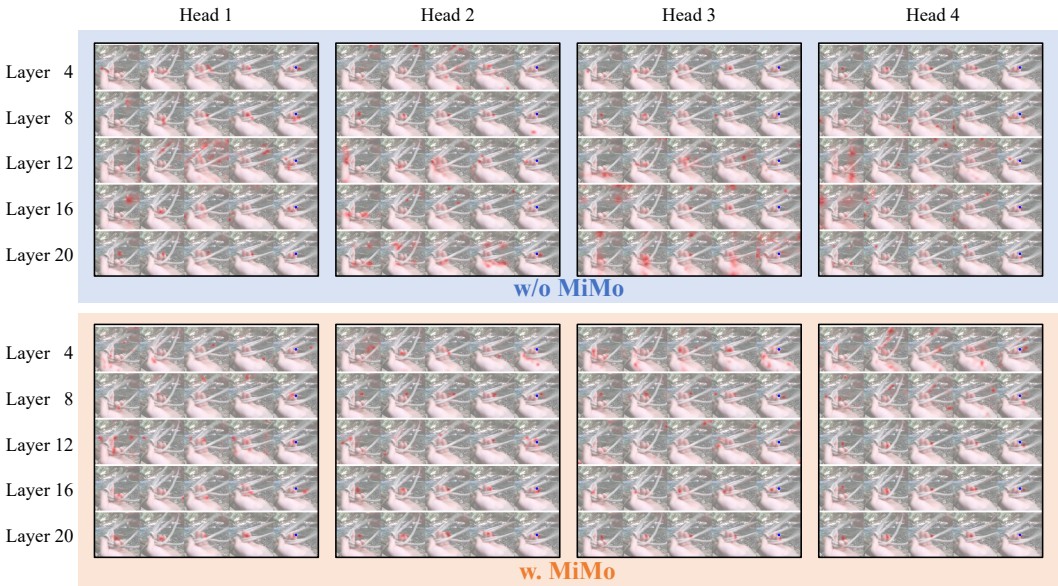

Figure 9: Attention heatmaps without and with MiMo.

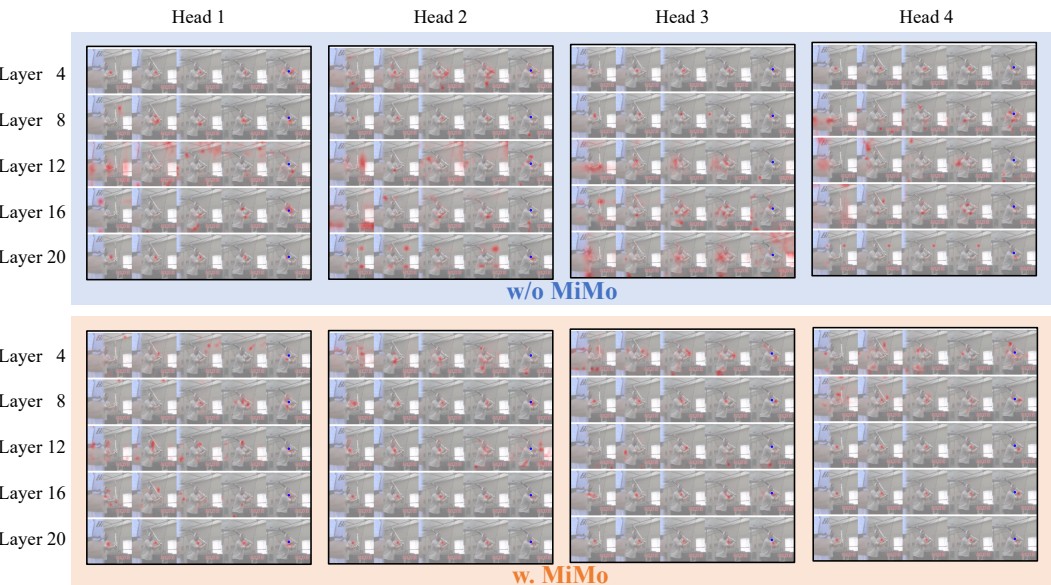

Figure 10: Attention heatmaps without and with MiMo.

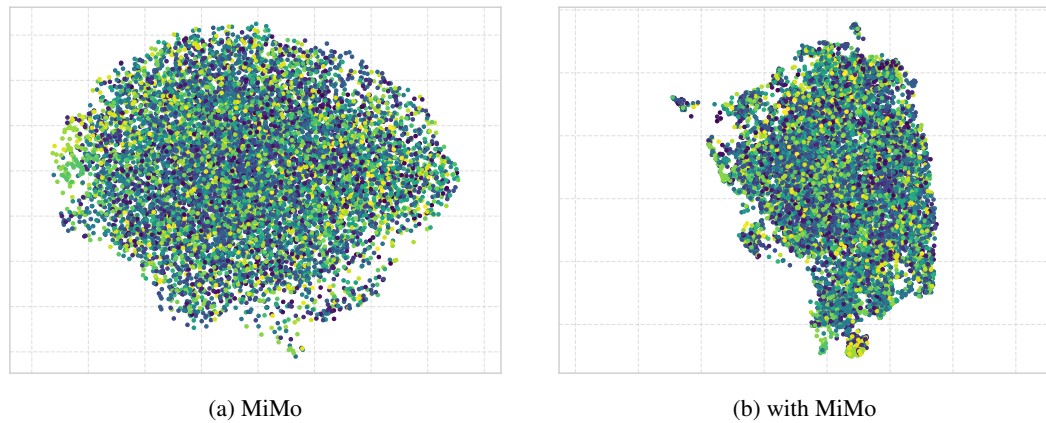

(a) MiMo            (b) with MiMo

Figure 11: UMAP visualization of video embeddings without and with MiMo.

the generated videos and the reference videos, using the strong V-JEPA 2 (Assran et al., 2025) pretrained model as the judge. It does not rely on the Gaussian assumption of the FVD metric, and gives more faithful evaluation (Jayasumana et al., 2024). Specifically, VMMD replaces the CLIP model in CMMD with V-JEPA 2 Large; other implementations are the same as in CMMD[4].

Table 11: Comparison of different methods with VMMD and FVD metrics.

| Method | ACDiT-XL | FAR-XL | MiMo-XL |
|--------|----------|--------|---------|
| FVD↓   | 10.264   | 9.311  | **8.257** |
| VMMD↓  | 1.075    | 1.036  | **0.977** |

Table 11 compares ACDiT, FAR, and MiMo with both VMMD and FVD metrics. The VMMD measurement results are consistent with the FVD, indicating that in our cases, FVD and VMMD can relatively well characterize the generation quality.

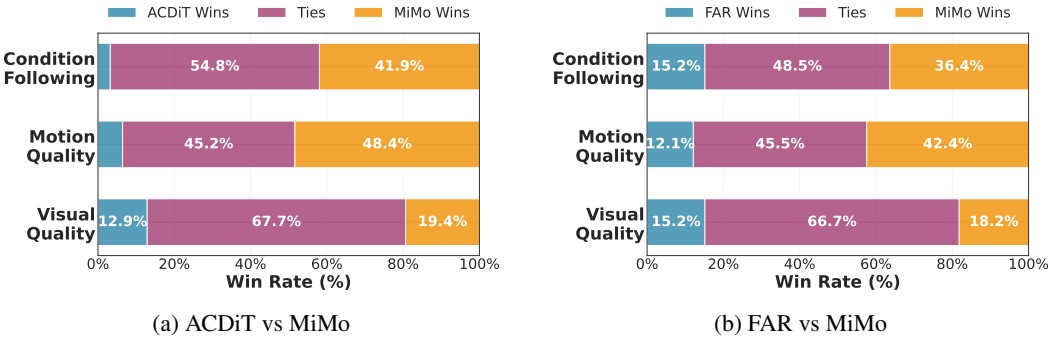

(a) ACDiT vs MiMo            (b) FAR vs MiMo

Figure 12: User studies (win rates) of ACDiT, FAR, and MiMo.

**User Studies** We conduct user studies to better understand what aspects MiMo improves. Five experts are instructed to evaluate 100 tasks, assessing three key dimensions: condition following, motion quality, and visual quality.

- Condition following: the visual and semantic consistency between conditioning frames and generated frames.
- Motion quality: whether there is motion distortion or motion that is semantically inconsistent with the context.

---

[4]https://github.com/google-research/google-research/tree/master/cmmd

- Visual quality: whether there is frame-level visual distortion or visual components that are semantically inconsistent with the context.

Figure 12 summarizes the results for ACDiT-XL, FAR-XL, and MiMo-XL, all trained for 360K steps on Kinetics-600. MiMo excels in condition following and motion quality, while the visual quality is marginally improved. Additionally, MiMo has higher win rates against ACDiT than against FAR, which indicates that lower VMMD and FVD metric values correlate with better perceptual quality in our cases.

## H.2 LONG-HORIZON VIDEO GENERATION

MiMo is effective for long-horizon video generation, as its robust history representation helps mitigate the error accumulation common in autoregressive models. We validate this on the action-conditioned Minecraft dataset (Yan et al., 2023), predicting 156 frames from 144, following the FAR (Gu et al., 2025) setup.

Table 12: Long-horizon video generation on the Minecraft dataset.

| Model | Steps | FVD↓ |
|---|---|---|
| FAR-B | 100K | 42.710 |
|  | 150K | 33.873 |
| MiMo-B | 100K | **33.829** |

The results in Table 12 show that MiMo significantly outperforms the baseline. At 100K training steps, MiMo achieves an rFVD of 33.829, a 27% improvement over the baseline's 42.710. This performance gap highlights MiMo's ability to maintain long-term coherence. Furthermore, MiMo accelerates training, reaching this performance 1.5x faster than the baseline, which requires an additional 50K steps to achieve a comparable rFVD. Figure 13 shows uncurated samples of generated results. These results confirm that a superior understanding of the past, enforced by MiMo, leads to more plausible and coherent long-term video generation.

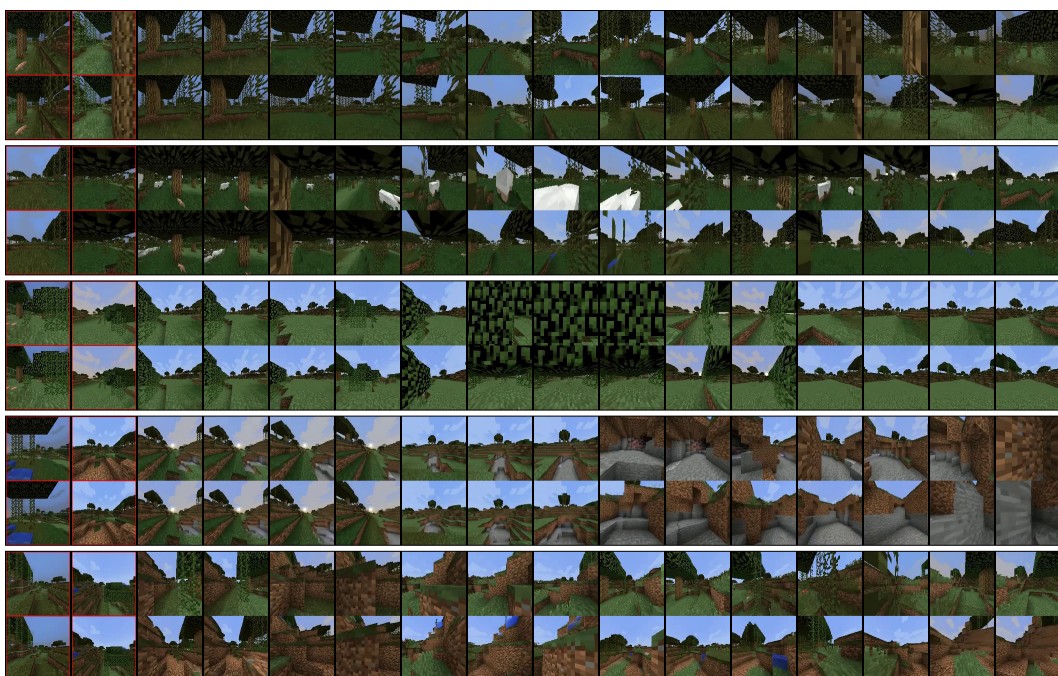

Figure 13: Uncurated samples of Minecraft long-horizon video generation. The upper row of each video is GT, the lower row is the generated sample. Red bounding boxes indicate conditioning frames.

## I  LIMITATIONS AND FUTURE WORK

In this work, we analyze the impact of the DiT's internal representations of history frames on VideoAR. Based on our findings, we propose MiMo to improve history representations *without* utilizing VFM.

However, it remains an open question to improve future frame representations with VFM. Masked DiT (Gao et al., 2023; Wei et al., 2023) achieves success to some extent but requires elaborate architecture modifications. Some recent approaches (Jiang et al., 2025; Wang & He, 2025) incorporate methodologies from self-supervised learning literature, but it is still unclear whether they (and MiMo) can beat representation alignment approaches (e.g., REPA (Yu et al., 2024)) that utilize pretrained VFM on in-distribution data of VFM. It is also unclear whether it is possible to pretrain a generative model that beats VFMs in downstream tasks such as video segmentation, video grounding, etc.

Furthermore, it is an interesting future direction to explore other training objectives to improve history representations (Oquab et al., 2023; Assran et al., 2023; Jiang et al., 2025; Wang & He, 2025).

