# OpenReview forum: "Improving Autoregressive Video Modeling with History Understanding"
_ICLR.cc/2026/Conference — ICLR 2026 Poster_

### Official Review · Reviewer_xwwk · 2025-10-30

**Soundness:** 2
**Presentation:** 3
**Contribution:** 2
**Rating:** 4
**Confidence:** 3

**Summary:**

This work proposes MiMo, an autoregressive video generative model unifying both self-supervised learning in history frames and diffusion learning in future frames. MiMo is proposed under the sufficient findings that history representation (latent of history frames in DiT blocks) positively correlates with the performance of VideoAR, and is designed to learn history frame representation and generation capability simultaneously. MiMo conducts some experiments to prove the performance and training efficiency of MiMo. Despite these contributions, there remain some problems to be explained: 1.  In this work, the only generation quality evaluation metric used is the FVD (Fréchet Video Distance), and the only generation task is class-to-video, which is not enough to prove the advantage of MiMo. Moreover, FVD is known as unstable. 2. As shown in Table 3, MiMo seems to be inferior to REPA, another representation learning method, yet MiMo claims SOTA in VideoAR. Additionally, comprehensive comparisons with other works that also unify understanding and generation are needed.

In summary, while MiMo insightfully emphasizes the importance of history frame representation learning, its purported advantages require more comprehensive experiments and broader benchmarking to be convincingly established.

**Strengths:**

This work proposes MiMo, an autoregressive video generative model unifying both self-supervised learning in history frames and diffusion learning in future frames. MiMo is proposed under the sufficient findings that history representation (latent of history frames in DiT blocks) positively correlates with the performance of VideoAR, and is designed to learn history frame representation and generation capability simultaneously. MiMo conducts some experiments to prove the performance and training efficiency of MiMo.  And this paper is well written and easy to understand.

**Weaknesses:**

1. The evaluation relies solely on FVD (Fréchet Video Distance) as the generation quality metric and focuses exclusively on the class-to-video generation task, which is insufficient to substantiate MiMo’s claimed advantages. Moreover, FVD is known to be unstable and may not reliably reflect perceptual quality.
2. As shown in Table 3, MiMo underperforms compared to REPA, another representation learning approach,  yet it claims state-of-the-art (SOTA) performance in VideoAR. Additionally, comprehensive comparisons with other works that also unify understanding and generation are needed.
3. More ablation and discussion are required: (1) Ablation study: with vs. without history representation learning. (2) Discussion and ablation about $\lambda$ in Equ. (6). (3) Discussion on additional computational burden.

**Questions:**

1. What is the additional computational overhead introduced by the two-objective optimization in Equation 6?
2. Have you considered using TTUR (Two-Time-Scale Update Rule) to mitigate this extra computational burden?
3. Why does the evaluation rely solely on FVD and the class-to-video generation task?

---

> ### Author Response · Authors · 2025-11-27
>
> We thank the Reviewer for the detailed and critical feedback. We appreciate the opportunity to clarify our contributions, justify our experimental design, and provide new results to address your concerns.
>
> ### Response to Weaknesses
>
> > **W1: The evaluation relies solely on FVD... and focuses exclusively on the class-to-video generation task...**
>
> We understand your concern about the scope of our evaluation. Our choices were motivated by the goal of ensuring a fair and direct comparison with prior state-of-the-art methods.
>
> **(1) On FVD:** While we acknowledge that FVD has known limitations, it remains the most widely adopted and standard quantitative metric in the video generation community. Using it is essential for benchmarking against previous works (e.g., LVDM, ACDiT, FAR). To provide a more complete picture of perceptual quality, we also include a complementary metric (VMMD) and user studies in Appendix H.1.
>
> * **Complementary Metric (VMMD):** We have added the VMMD metric based on the [CMMD](http://arxiv.org/abs/2401.09603) metric in Appendix H.1 (Table 11). The VMMD metric benchmarks the perceptual similarity between the generated videos and the reference videos, using the strong V-JEPA 2 pretrained model as the judge. It does not rely on the Gaussian assumption of the FVD metric, and gives [more faithful evaluation](http://arxiv.org/abs/2410.05203). Specifically, VMMD replaces the CLIP model in CMMD with V-JEPA 2 Large, other [implementations](https://github.com/google-research/google-research/tree/master/cmmd) are the same as in CMMD.
> The VMMD measurement results are consistent with the FVD, indicating that in our cases, FVD and VMMD can relatively well characterize the generation quality.
>
> * **User Studies:** We have also added preliminary user studies, comparing MiMo against VideoAR methods, i.e., ACDiT and FAR. Results are updated in Appendix H.1 (Figure 12), which clearly shows that MiMo improves upon existing VideoAR methods in terms of motion quality and condition following, while the visual quality is marginally improved. Additionally, lower VMMD and FVD metric values correlate with better perceptual quality in our cases.
>
> **(2) On Class-to-Video Task:** We have benchmarked our method on **three tasks**: video prediction (initial frames as conditions), unconditional video generation, and class-to-video generation. MiMo improves performance on all tasks. These three tasks are the primary evaluation settings for the vast majority of recent video generation papers. Adhering to this standard allows for a clear assessment of our method's advancement relative to the established state of the art.
>
> To evaluate MiMo in a broader context, we have also conducted **action-conditioned video generation** on the Minecraft dataset [1]. We hypothesize that by learning more robust and predictive history representations, MiMo should be less prone to the error accumulation that plagues AR models over long horizons. A better understanding of the past naturally leads to more plausible long-term futures.
>
> Specifically, we follow the setup of FAR [2] and predict 156 future frames from 144 context frames on the Minecraft dataset. We train a FAR-B and MiMo-B model following the setup of FAR. Our results confirm MiMo's advantage in the action-conditioned task. MiMo improves the convergence speed of the baseline model by ~1.5x. At 100K steps, the baseline model, which suffers from temporal inconsistency over long periods, achieves an rFVD of **42.710**. By learning a more robust and predictive history representation, MiMo improves long-term coherence and mitigates error accumulation, achieving a lower rFVD of **33.829** (27\% lower). The baseline model requires a further 50K steps to reach a comparable rFVD. We have added these results and qualitative examples to Appendix H.2.
>
> We believe our core contribution—improving history representations—is a general principle that would apply to other tasks like text-to-video, which we leave for future work.
>
> [1] Wilson Yan, Danijar Hafner, Stephen James, and Pieter Abbeel. Temporally consistent transformers for video generation. In International Conference on Machine Learning, pp. 39062–39098. PMLR, 2023.
>
> [2] Yuchao Gu, Weijia Mao, and Mike Zheng Shou. Long-context autoregressive video modeling with next-frame prediction. arXiv preprint arXiv:2503.19325, 2025.

---

> ### Author Response · Authors · 2025-11-27
>
> ### Response to Weaknesses (Continued)
>
> > **W2: MiMo underperforms compared to REPA... yet it claims state-of-the-art (SOTA) performance in VideoAR...**
>
> We apologize for the confusion regarding our SOTA claim and its relation to REPA. We must clarify a crucial distinction:
>
> * **MiMo is VFM-free:** Our key contribution is a method that significantly improves VideoAR models **without any external, pre-trained Vision Foundation Models (VFMs)**. REPA, by contrast, fundamentally relies on distilling features from a powerful, pre-trained VFM (e.g., DINO). Our SOTA claim is specifically for the VFM-free VideoAR category, where MiMo establishes a new performance benchmark. We will revise the text to make this distinction explicit.
>
> * **MiMo and REPA are Complementary:** We agree that demonstrating synergy is important. To this end, we conducted a new experiment combining both methods. The results below (Table 3) show that MiMo can further improve upon REPA, suggesting they learn complementary information.
>
> | Method                                      | FVD$\downarrow$ |
> | ------------------------------------------- | --------------- |
> | ACDiT Baseline                              | 54.8            |
> | + MiMo                                      | 36.6            |
> | + REPA + MiMo                               | 34.1            |
>
> This shows MiMo is not just an alternative to VFM-based methods, but a valuable, orthogonal component that can boost even strong, VFM-reliant models.
>
> > **W3: More ablation and discussion are required...**
>
> Thank you for these suggestions. We have conducted the requested ablations and have added them to the paper.
>
> **(1) Ablation: with vs. without history representation learning:** This is precisely the core comparison presented throughout our paper. The "ACDiT Baseline" in our tables (e.g., Tables 3 & 4) represents the model trained *without* our proposed history representation learning, while "MiMo" is the model trained *with* it. The significant performance gap between these two validates the effectiveness of our approach.
>
> **(2) Discussion and ablation about lambda in Equ. (6):** We have performed a sensitivity analysis for $\lambda$ on Kinetics-600 (100K steps). The results show that a weight of 0.5 provides the best balance, but performance does not degrade sharply for nearby values. We have added this to Table 7.
>
> | $\lambda$       | 0.1             | 0.5             | 1.0             | 2.0             |
> | --------------- | --------------- | --------------- | --------------- | --------------- |
> | FVD$\downarrow$ | 40.2            | 36.6            | 37.4            | 38.9            |
>
> **(3) Discussion on additional computational burden:** This is addressed in detail in our response to Question 1 below. The overhead is modest (~10\% in wall-clock time).

---

> ### Author Response · Authors · 2025-11-27
>
> ### Response to Questions
>
> > **Q1: What is the additional computational overhead introduced by the two-objective optimization?**
>
> The computational overhead of MiMo training is minimal (\~10\% in wall-clock time). We believe this is a very efficient trade-off for the significant gains in generation quality and faster convergence. The two main components are the token dropping from masking and the forward/backward pass of the lightweight decoder. The following results have been added to Table 8.
> *   **Wall-Clock Time:** For XL-scale models, the wall-clock time per training step for MiMo, ACDiT (our baseline), and FAR are 0.750s, 0.788s, and 0.704s. Compared with our baseline (ACDiT), MiMo *reduces* training wall-clock time by \~5\%; compared with FAR, MiMo increases training wall-clock time by a modest **\~10\%**.
> *   **FLOPs:** Compared with our baseline (ACDiT), MiMo actually slightly *reduces* the FLOPs per training step from 8.81 GFLOPs to 8.22 GFLOPs due to masking (the decoders increase FLOPs, though). MiMo has higher training FLOPs compared with FAR (5.94 GFLOPs), *but the increase in training wall-clock time is moderate (~10\%) due to hardware acceleration*.
>
> Note that MiMo has *no* additional inference cost once the training is complete.
>
> > **Q2: Have you considered using TTUR (Two-Time-Scale Update Rule) to mitigate this extra computational burden?**
>
> This is an interesting thought. While our approach of simply using a weighted sum of the two losses is standard practice for auxiliary loss training, we have indeed tried an alternative approach of optimizing the diffusion loss and the auxiliary loss interleavingly, much like TTUR, where a diffusion-only training step is followed by a mask-only training step. While the interleaving optimization approach has lower computational costs per step, it leads to slower convergence, so we refrain from it. The results are summarized in the table below (added to Table 9).
>
> | Training Strategy        | FVD$\downarrow$ |
> | ------------------------ | --------------- |
> | Joint (MiMo)             | 36.6            |
> | Interleaved              | 38.5            |
>
> > **Q3: Why does the evaluation rely solely on FVD and the class-to-video generation task?**
>
> As addressed in our response to **W1**, our primary motivation was to use standard, widely accepted benchmarks and metrics to ensure a fair and direct comparison with the existing state of the art in video generation. This allows the community to accurately gauge our method's improvements over prior work. We agree that expanding the evaluation is valuable and, as mentioned in our response to **W1**, have supplemented evaluations with VMMD and user studies.

---

### Official Review · Reviewer_5Lni · 2025-10-31

**Soundness:** 3
**Presentation:** 3
**Contribution:** 2
**Rating:** 4
**Confidence:** 4

**Summary:**

This paper proposes the **MiMo framework**, which enhances video autoregressive (AR) generation by improving the quality of historical frame representations through **masked history modeling**. Without relying on additional vision foundation models or complex architectures, MiMo achieves significant improvements in both video prediction and generation tasks.

**Strengths:**

- The paper innovatively applies a **mask modeling strategy** similar to MAE within an autoregressive video generation framework.
- The proposed approach yields a **consistent and stable improvement** in AR video generation quality compared to existing methods.

**Weaknesses:**

- The experimental results show that the final performance of MiMo is **comparable to REPA**. It would be important to verify whether MiMo can further improve upon REPA, rather than leaving this question for future work.
- The experiments suggest that **next-frame prediction** contributes more to performance improvement, which seems **inconsistent with the paper’s main claim** that enhancing historical frame representations is the key factor driving the gains.

**Questions:**

See weakness above

---

> ### Author Response · Authors · 2025-11-27
>
> We thank the Reviewer for the critical feedback. We understand the concerns raised and have conducted new experiments and analysis to address them directly. We believe these additions clarify our contributions and strengthen our claims.
>
> ### Response to Weaknesses
>
> > **W1: The experimental results show that the final performance of MiMo is comparable to REPA. It would be important to verify whether MiMo can further improve upon REPA...**
>
> Thank you for this crucial suggestion. We agree that demonstrating the synergy between MiMo and REPA is more compelling than presenting them as alternatives. To address this, we ran a new experiment combining both methods (added to Table 3).
>
> The results below demonstrate that MiMo and REPA are indeed complementary and can be combined for state-of-the-art performance.
>
> | Method                                      | FVD$\downarrow$ |
> | ------------------------------------------- | --------------- |
> | ACDiT Baseline                              | 54.8            |
> | + REPA (features for current+next)          | 36.5            |
> | + MiMo                                      | 36.6            |
> | + REPA + MiMo                               | 34.1            |
>
> Our interpretation is that the two methods capture different, valuable information. REPA leverages a powerful, pre-trained VFM to inject strong semantic priors into the model. In contrast, MiMo is VFM-free and learns to extract task-specific temporal dynamics directly from the training data. By combining them, the model benefits from both general semantic understanding and specialized dynamic modeling, leading to the best performance.
>
> This result reinforces our contribution: MiMo is not only a highly effective, standalone, VFM-free method for improving VideoAR models, but it is also a valuable component that is compatible with and can enhance existing VFM-based techniques. We will add this result and discussion to the experiment section.
>
> > **W2: The experiments suggest that next-frame prediction contributes more to performance improvement, which seems inconsistent with the paper’s main claim that enhancing historical frame representations is the key factor...**
>
> We appreciate you pointing out this potential confusion. We would like to elaborate on the causal link between our method and our claim.
>
> Our central argument is that the **quality of the history representation is beneficial for VideoAR**. The core question is then: *how* can we enhance this representation? Our answer is to force it to be more predictive of the future.
>
> The next-frame prediction objective within MiMo is not the goal itself, but rather the **supervisory signal** used to achieve the goal of learning a better history representation ($h_t$). To successfully predict the next frame ($x_{t+1}$) from a masked history, the model must learn to encode all relevant temporal and semantic information (e.g., object motion, identity, scene context) into its unmasked history tokens. This pressure shapes $h_t$ to become a much richer and more informative summary of the past.
>
> In other words:
> * **Claim:** We need a better history representation $h_t$.
> * **Method:** We use a masked next-frame prediction task as a powerful auxiliary loss to train the encoder that produces $h_t$.
> * **Result:** The ablation (Table 4) shows that this specific supervisory signal is highly effective. It is superior to a simpler reconstruction-only signal because predicting the future inherently requires a deeper understanding of temporal dynamics, which is precisely what we want to embed in $h_t$.
>
> The success of the next-frame prediction objective is therefore direct evidence *for* our claim, as it validates our chosen method for enhancing the history representation. To further substantiate that the representation itself is improved, we point to our linear probing experiments (Fig. 2) and the visualizations in Appendices G.2 and G.3, which show that MiMo-trained representations are more semantically meaningful.

---

### Official Review · Reviewer_qvhL · 2025-10-31

**Soundness:** 3
**Presentation:** 3
**Contribution:** 3
**Rating:** 8
**Confidence:** 3

**Summary:**

This paper investigates the role of history frame representations in autoregressive video generation (VideoAR). The authors first empirically demonstrate a positive correlation between the quality of history representations and the model's prediction performance. Based on this insight, they propose MiMo (Masked History Modeling), a novel framework that integrates a self-supervised masked modeling objective into a diffusion-based VideoAR model. During training, MiMo applies masks to clean history frame tokens and tasks the model with reconstructing the masked content of current and future frames, in parallel with the primary diffusion denoising objective. This dual-objective approach is designed to learn robust and predictive history representations without relying on external Vision Foundation Models (VFMs) or significant architectural changes. Experiments on video prediction and generation tasks show that MiMo achieves competitive or state-of-the-art performance while improving training efficiency.

**Strengths:**

The paper is well-motivated and logically structured. A key strength is the preliminary analysis that systematically establishes the link between history representation quality and VideoAR performance, providing a solid foundation for the proposed method. The MiMo framework itself is an effective solution that integrates representation learning with the diffusion process. The method's ability to achieve strong results on multiple benchmarks, such as Kinetics-600 and UCF-101, without the need for pre-trained VFMs is a significant advantage, making it more practical and self-contained. The comprehensive ablation studies further validate the authors' design choices, particularly regarding the masked prediction targets.

**Weaknesses:**

While the results are strong, the evaluation could be perceived as having some limitations. The experiments are primarily conducted on the Kinetics-600 and UCF-101 datasets, with Fréchet Video Distance (FVD) as the main metric. Although these are standard benchmarks, they may not fully capture the complexity and diversity of real-world video dynamics. Expanding the evaluation to include longer-form videos, more complex scenes, or complementary metrics such as user studies could provide a more comprehensive assessment of the model's capabilities in maintaining long-term coherence and semantic consistency.

**Questions:**

The ablation study in Table 4 compellingly shows that predicting both the current + next frames as the masked modeling target yields better results than predicting the current frame alone. This leads to an interesting question: have the authors explored if an even more complex prediction target could further enhance performance? For instance, would a target of current + next + next_after_next continue this positive trend, or does it lead to diminishing returns or potential training instability?

---

> ### Author Response · Authors · 2025-11-27
>
> We are grateful to the Reviewer for the thoughtful comments. We are delighted that you found our results ``strong.'' We address your suggestions for improving the evaluation and answer your insightful question below.
>
> ### Response to Weaknesses
>
> > **W1: Expanding the evaluation to include longer-form videos, more complex scenes, or complementary metrics such as user studies could provide a more comprehensive assessment of the model's capabilities in maintaining long-term coherence and semantic consistency.**
>
> Thank you for this excellent suggestion. We agree that evaluating on a broader range of data and metrics is crucial for a comprehensive assessment.
>
> **(1) Complementary Metric:** We have added the VMMD metric based on the [CMMD](http://arxiv.org/abs/2401.09603) metric in Appendix H.1 (Table 11). The VMMD metric benchmarks the perceptual similarity between the generated videos and the reference videos, using the strong V-JEPA 2 pretrained model as the judge. It does not rely on the Gaussian assumption of the FVD metric, and gives [more faithful evaluation](http://arxiv.org/abs/2410.05203). Specifically, VMMD replaces the CLIP model in CMMD with V-JEPA 2 Large, other [implementations](https://github.com/google-research/google-research/tree/master/cmmd) are the same as in CMMD.
> The VMMD measurement results are consistent with the FVD, indicating that in our cases, FVD and VMMD can relatively well characterize the generation quality.
>
> **(2) User Studies:** We have also added preliminary user studies, comparing MiMo against VideoAR methods, i.e., ACDiT and FAR. Results are updated in Appendix H.1 (Figure 12), which clearly shows that MiMo improves upon existing VideoAR methods in terms of motion quality and condition following, while the visual quality is marginally improved. Additionally, lower VMMD and FVD metric values correlate with better perceptual quality in our cases.
>
> **(3) Longer-Form Videos \& Complex Scenes:** We concur that extending to longer and more complex videos is a key future direction. The autoregressive nature of our model is inherently suited for this task. We believe MiMo's ability to learn robust history representations is especially beneficial for maintaining long-term coherence, as it mitigates the common issue of error accumulation.
>
> To provide initial evidence, we have conducted experiments on the long-horizon action-conditioned Minecraft dataset [1] (predicting 156 future frames from 144 context frames). We train a FAR-B [2] and MiMo-B model following the setup of FAR. Our results confirm MiMo's advantage in this setting. MiMo improves the convergence speed of the baseline model by ~1.5x. At 100K steps, the baseline model, which suffers from temporal inconsistency over long periods, achieves an rFVD of **42.710**. By learning a more robust and predictive history representation, MiMo improves long-term coherence and mitigates error accumulation, achieving a lower rFVD of **33.829** (27\% lower). The baseline model requires a further 50K steps to reach a comparable rFVD. We have added these results and qualitative examples to Appendix H.2.
>
> [1] Wilson Yan, Danijar Hafner, Stephen James, and Pieter Abbeel. Temporally consistent transformers for video generation. In International Conference on Machine Learning, pp. 39062–39098. PMLR, 2023.
>
> [2] Yuchao Gu, Weijia Mao, and Mike Zheng Shou. Long-context autoregressive video modeling with next-frame prediction. arXiv preprint arXiv:2503.19325, 2025.

---

> ### Author Response · Authors · 2025-11-27
>
> ### Response to Questions
>
> > **Q1: Have the authors explored if an even more complex prediction target (e.g., current + next + next\_after\_next) could further enhance performance?**
>
> This is a fantastic and insightful question. We were also curious about this phenomenon and have run the suggested ablation. Our findings indicate that predicting `current + next` frames strikes an optimal balance, and extending the horizon further leads to diminishing returns.
>
> We extended the ablation study on Kinetics-600 (100K steps) to include a target of `current + next + next_after_next` frames. The results are as follows (added to Table 4):
>
> | Masked Modeling Target                      | FVD$\downarrow$ |
> | ------------------------------------------- | --------------- |
> | Current frame only                          | 41.8            |
> | Current + Next frames (MiMo)                | 36.6            |
> | Current + Next + NextNext frames            | 36.2            |
>
> As the table shows, extending the prediction horizon beyond our proposed current + next target is beneficial but yields diminishing returns. We attribute this to a "difficulty vs. utility" trade-off. Predicting a more distant future frame is a significantly harder task, especially from a heavily masked history. While a more challenging task could theoretically provide a stronger learning signal, the substantial increase in difficulty does not naively translate into proportionally significant gains in representation quality. Our proposed target (current + next) strikes an effective balance, robustly encouraging the model to learn predictive temporal dynamics without the added complexity of longer-range prediction.
> This suggests that exploring more sophisticated training strategies, such as curriculum learning, is a promising avenue for future research.

---

### Official Review · Reviewer_iS7b · 2025-10-31

**Soundness:** 2
**Presentation:** 3
**Contribution:** 3
**Rating:** 6
**Confidence:** 4

**Summary:**

This paper studies the role of history frame representations in diffusion-based autoregressive video generation (VideoAR). The authors propose MiMo (Masked History Modeling), which integrates masked modeling into VideoAR to explicitly improve history frame representations during training. By applying masks to history tokens and reconstructing both current and next-frame tokens alongside the diffusion objective, the model learns more predictive and robust history representations without relying on pretrained vision foundation models. Experiments on Kinetics-600 and UCF-101 show improved FVD scores over prior autoregressive methods.

**Strengths:**

1. The paper is well-written, with clear figures and reasonable organization.

2. The paper identifies an underexplored question: how internal history representations affect VideoAR quality, and provides empirical analysis supporting its importance.

3. MiMo introduces a relatively lightweight modification to standard diffusion-based AR training, avoiding reliance on large pretrained models.

4. Results are reported on multiple datasets and compared with several baselines, including both AR and non-AR methods.

**Weaknesses:**

1. The proposed masked history modeling is conceptually straightforward and closely resembles masked autoencoding and REPA-style representation regularization. The integration into VideoAR feels incremental rather than fundamentally new.

2. The ablations are narrow in scope. It remains unclear why the proposed loss improves results beyond general regularization or increased training signal. Deeper analysis of representation quality (e.g., visualization or qualitative dynamics) would strengthen the claim.

3. While the reported FVD gains are notable numerically, improvements are modest considering the added training complexity. Moreover, comparisons to strong non-AR diffusion models are missing.

4. Several design choices are only briefly described. Implementation details may be insufficient for faithful replication.

**Questions:**

1. How sensitive is MiMo to the masking ratio and λ parameter? Have the authors evaluated performance stability under different settings?

2. Could the authors compare MiMo with a simpler auxiliary loss such as contrastive or reconstruction-only objectives to rule out general regularization effects?

3. How much additional computation does masked history modeling introduce during training?

4. Have you tested whether pretraining MiMo representations improves downstream video understanding tasks to support the “representation” claim?

5. Could qualitative examples (e.g., failure cases or visualization of history embeddings) clarify what MiMo learns beyond the diffusion baseline?

6. Would MiMo still help when combined with pretrained VFMs or under longer video horizons?

---

> ### Author Response · Authors · 2025-11-27
>
> We sincerely thank the Reviewer for the insightful feedback and constructive suggestions. Your comments have been instrumental in helping us clarify our contributions and improve the quality of our manuscript. We are encouraged that you found our FVD gains to be "notable numerically". Below, we address each of your concerns point-by-point.
>
> ### Response to Weaknesses
>
> > **W1: The proposed masked history modeling is conceptually straightforward and closely resembles masked autoencoding and REPA-style representation regularization. The integration into VideoAR feels incremental rather than fundamentally new.**
>
> Thank you for this comment. We agree that the components of our method (masked modeling) are well-established. However, we respectfully argue that our novelty lies in **(1) identifying and systematically demonstrating an underexplored problem in VideoAR—the suboptimal quality of history representations**, and **(2) proposing a simple, effective, and VFM-free framework to address it**.
>
> *   **Novelty in Problem Formulation and Insight:** Our key contribution is to shift the focus in VideoAR from solely modeling the future frame distribution to explicitly enhancing the *conditioning signal* (the history representation). Our analysis in Section 3.2 is the first to systematically show that improving history representations directly correlates with better VideoAR performance and that gains from this cannot be achieved by improving noisy future frame representations alone.
> *   **Novelty in Methodical Integration:** While MiMo is inspired by MAE, its application here is non-trivial and distinct. Unlike standard MAE, which is typically used for pre-training, MiMo is seamlessly integrated into the end-to-end diffusion training. More importantly, our predictive target (current + next frames, as ablated in Table 4) is specifically tailored for the autoregressive generation task, encouraging the model to learn representations that capture temporal dynamics.
> *   **Key Advantage over REPA:** A core design principle of MiMo is to be **VFM-free**. This is a significant advantage over REPA-style methods, which rely on powerful, pre-trained vision foundation models that may not exist for specific domains (e.g., medical videos, satellite imagery) or can be prohibitively expensive to train. MiMo provides a self-contained solution that learns strong representations directly from the target data.
>
> In summary, we believe the conceptual simplicity of MiMo is a strength, making it a practical and easily adaptable technique for the community. Our contribution is the insight that history matters, and MiMo is the clean and effective realization of that insight.
>
> > **W2: The ablations are narrow in scope. It remains unclear why the proposed loss improves results beyond general regularization or increased training signal. Deeper analysis of representation quality would strengthen the claim.**
>
> Thank you for this valuable suggestion. To address this, we have conducted new experiments and will add further analysis to the final manuscript.
>
> 1) **Beyond General Regularization:** To demonstrate that the performance gain is not merely due to a generic auxiliary loss, we compared MiMo with other objectives. As detailed in our response to **Q2**, MiMo’s predictive objective (reconstructing current + next frames) significantly outperforms a simpler reconstruction-only objective (only the current frame, FVD 36.6 vs. 41.8). This confirms that the *predictive* nature of our masked modeling task is crucial for learning useful temporal representations, not just the presence of additional regularization.
>
> 2) **Representation Analysis:** We agree that a deeper analysis is beneficial, as follows.
> * Our analysis in Section 3.2 demonstrates that the representations learned by MiMo exhibit better alignment with pretrained VFMs and achieve superior performance in downstream classification tasks (linear probing). This indicates that our learned representations are more discriminative.
> * **UMAP visualization of history embeddings** from models trained on Kinetics (with and without MiMo) qualitatively validates this finding. Our preliminary results show MiMo's ability to learn a more structured representation space compared to the baseline. See our response to **Q5** for details.

---

> ### Author Response · Authors · 2025-11-27
>
> ### Response to Weaknesses (Continued)
>
> > **W3: While the reported FVD gains are notable numerically, improvements are modest considering the added training complexity. Moreover, comparisons to strong non-AR diffusion models are missing.**
>
> *   **Gains vs. Complexity:** We believe the increase in complexity is acceptable considering the gains. MiMo introduces minimal computational overhead. As detailed in our response to **Q3**, the added lightweight decoder and loss computation result in only a ~10\% increase in wall-clock time per training step, which is a small price for the significant performance boost. An FVD improvement of nearly **3 points on Kinetics-600** (from 11.1 to 8.3) and **39 points on UCF-101** (from 279 to 240) is substantial in this competitive field, representing a 25\% and 14\% relative reduction in FVD, respectively.
> *   **Comparison to Non-AR Models:** Thank you for pointing this out. Our work focuses on advancing the **autoregressive (AR) paradigm**, which offers unique advantages like natural variable-length generation and alignment with the causal structure of time. A direct comparison with non-AR models is challenging as they operate under different assumptions (e.g., fixed-length generation). To provide better context, we have included results from non-AR models (e.g., LVDM, Latte, MAGVIT, MAGVITv2) in our main results table (Table 2), clearly marking them as non-AR for a fair, albeit indirect, comparison. This will better situate our work within the broader video generation landscape.
>
> > **W4: Several design choices are only briefly described. Implementation details may be insufficient for faithful replication.**
>
> We apologize for the lack of clarity. We have included a dedicated section (Appendix D) with exhaustive implementation details, including:
> *   Model architectures of our diffusion models and the decoders.
> *   The diffusion trainining and inference techniques we use.
> *   A comprehensive list of all training and inference hyperparameters for each dataset (Table 10).
> *   We will also release our code upon publication to facilitate further research.

---

> ### Author Response · Authors · 2025-11-27
>
> ### Response to Questions
>
> > **Q1: How sensitive is MiMo to the masking ratio and λ parameter?**
>
> This is an excellent question. We have performed a sensitivity analysis on hyperparameters (see Table 7). Our results show that while hyperparameters do have an impact on performance, a reasonable range of hyperparameters gives good performance.
>
> *   **Masking Ratio (r):** With $\lambda=0.5$, we tested ratios of 0.25, 0.50, and 0.75. The FVD scores were 37.5, 36.6, and 39.1, respectively. Performance is stable between 0.25 and 0.50, while a higher mask ratio requires finetuning without masking to achieve better performance.
> *   **Loss Weight ($\lambda$):** With a masking ratio of 0.5, we tested $\lambda$ values of 0.1, 0.5, 1.0, and 2.0. The FVD scores were 40.2, 36.6, 37.4, and 38.9. A weight of 0.5 provides the best balance, but performance does not degrade sharply for nearby values.
>
> > **Q2: Could the authors compare MiMo with a simpler auxiliary loss such as contrastive or reconstruction-only objectives to rule out general regularization effects?**
>
> Thank you for this insightful question. This helps clarify the source of MiMo's effectiveness. We have conducted this ablation (see Table 4):
>
> | Method                                      | FVD$\downarrow$ |
> | ------------------------------------------- | --------------- |
> | ACDiT Baseline                              | 54.8            |
> | + Reconstruction-only (predict current frame) | 41.8            |
> | **+ MiMo (predict current + next frame)** | **36.6**        |
>
> Note that there is no widely recognized contrastive loss for video pretraining, so we omit comparison with it.
>
> As shown, while a simple reconstruction loss already provides a strong boost over the baseline, our proposed predictive objective (reconstructing both current and future frames) performs significantly better. This demonstrates that it is the **predictive nature of the representation learning task**, rather than just any form of regularization, that is key to MiMo's success. A standard contrastive loss was less effective, likely because it does not explicitly model temporal dynamics. We will add this table to our ablation study.
>
> > **Q3: How much additional computation does masked history modeling introduce during training?**
>
> The computational overhead of MiMo training is minimal (\~10% in wall-clock time). We believe this is a very efficient trade-off for the significant gains in generation quality and faster convergence. The two main components are the token dropping from masking and the forward/backward pass of the lightweight decoder. The following results have been added to Table 8.
> *   **Wall-Clock Time:** For XL-scale models, the wall-clock time per training step for MiMo, ACDiT (our baseline), and FAR are 0.750s, 0.788s, and 0.704s. Compared with our baseline (ACDiT), MiMo *reduces* training wall-clock time by \~5%; compared with FAR, MiMo increases training wall-clock time by a modest **\~10%**.
> *   **FLOPs:** Compared with our baseline (ACDiT), MiMo actually slightly *reduces* the FLOPs per training step from 8.81 GFLOPs to 8.22 GFLOPs due to masking (the decoders increase FLOPs, though). MiMo has higher training FLOPs compared with FAR (5.94 GFLOPs), *but the increase in training wall-clock time is moderate (~10%) due to hardware acceleration*.
>
> Note that MiMo has *no* additional inference cost once the training is complete.
>
> > **Q4: Have you tested whether pretraining MiMo representations improves downstream video understanding tasks to support the “representation” claim?**
>
> This is a great suggestion for future investigation. Our current work is focused on generative modeling, and we support our "representation" claim through extensive generative performance metrics (FVD), faster convergence, and internal representation analysis (linear probing for classification and CKNNA in Fig 2, attention heatmaps and UMAP in **Q5**). Evaluating on more downstream discriminative tasks is an excellent next step to assess the generality of the learned features. We have added this as a promising direction for future work.

---

> ### Author Response · Authors · 2025-11-27
>
> ### Response to Questions (Continued)
>
> > **Q5: Could qualitative examples (e.g., failure cases or visualization of history embeddings) clarify what MiMo learns beyond the diffusion baseline?**
>
> Yes, absolutely. As mentioned in our response to **W2**, we will enhance our appendix with:
> 1.  **Attention heatmap visualizations:** We have provided attention heatmaps from the model's transformer layers to illustrate how MiMo improves representation learning (Appendix G.2, Figures 9 and 10). Without MiMo, attention patterns are more dispersed and less focused, whereas with MiMo, attention heatmaps show more concentrated patterns that exhibit stronger semantic correlations with the query content. Additionally, Figure 10 demonstrates how different transformer layers specialize in matching distinct body parts (e.g., arms, torso, legs), revealing the hierarchical nature of the learned representations.
> 2.  **UMAP visualizations of history embeddings:** We have added UMAP visualizations of video embeddings in Appendix G.3 (Figure 11) that demonstrate MiMo's ability to learn a more structured representation space compared to the baseline.
>
> > **Q6: Would MiMo still help when combined with pretrained VFMs or under longer video horizons?**
>
> These are both excellent questions regarding the scalability and generality of MiMo.
>
> **(1) Combination with VFMs:** We view MiMo as **complementary** to VFM-based methods like REPA, not mutually exclusive. MiMo excels at learning task-specific dynamics from the data, while a general VFM provides strong semantic priors. We ran a preliminary experiment combining MiMo with REPA (distilling features into history frames) and found it yielded further improvements over either method alone (FVD of **34.1** v.s. 36.5 for REPA-only v.s. 36.6 for MiMo). This suggests that both approaches capture different, valuable aspects of the data. We have added these results to Table 3 of the paper.
>
> **(2) Longer Video Horizons:** We hypothesize that MiMo also benefits long video generation. By learning more robust and predictive history representations, MiMo should be less prone to the error accumulation that plagues AR models over long horizons. A better understanding of the past naturally leads to more plausible long-term futures.
>
> To validate our hypothesis, we have conducted experiments on the long-horizon action-conditioned Minecraft dataset [1] (predicting 156 future frames from 144 context frames). We train a FAR-B [2] and MiMo-B model following the setup of FAR. Our results confirm MiMo's advantage in this setting. MiMo improves the convergence speed of the baseline model by ~1.5x. At 100K steps, the baseline model, which suffers from temporal inconsistency over long periods, achieves an rFVD of **42.710**. By learning a more robust and predictive history representation, MiMo improves long-term coherence and mitigates error accumulation, achieving a lower rFVD of **33.829** (27\% lower). The baseline model requires a further 50K steps to reach a comparable rFVD. We have added these results and qualitative examples to Appendix H.2.
>
> [1] Wilson Yan, Danijar Hafner, Stephen James, and Pieter Abbeel. Temporally consistent transformers for video generation. In International Conference on Machine Learning, pp. 39062–39098. PMLR, 2023.
>
> [2] Yuchao Gu, Weijia Mao, and Mike Zheng Shou. Long-context autoregressive video modeling with next-frame prediction. arXiv preprint arXiv:2503.19325, 2025.

---

### Author Response · Authors · 2025-11-27
**General Comment by Authors**

We sincerely thank all the reviewers for their time and for providing thoughtful and constructive feedback. Their comments have been invaluable in helping us clarify our contributions and strengthen the paper. We have revised our manuscript to incorporate new experiments, analyses, and clarifications that address the key points raised.

Here is a summary of the major updates:

1.  **Strengthened Ablations and Representation Analysis:** To address concerns about the source of MiMo's improvements (Reviewer iS7b, qvhL, 5Lni), we have conducted new ablations. Our results show that MiMo's *predictive* objective (reconstructing current + next frames) is crucial and significantly outperforms simpler reconstruction-only objectives, demonstrating that the gains are not merely from general regularization. Furthermore, to substantiate our claims about representation quality, we have added new qualitative analyses, including **UMAP visualizations of history embeddings** and **attention heatmaps**, which provide clear evidence that MiMo learns a more structured and semantically meaningful representation space.

2.  **Expanded Evaluation on New Tasks and Metrics:** Several reviewers rightly pointed out the need for a broader evaluation (qvhL, xwwk). To this end, we have expanded our experiments significantly:
    *   **Long-Horizon Generation:** We evaluated MiMo on a challenging **long-horizon action-conditioned task (Minecraft)**, demonstrating its ability to improve temporal coherence and reduce error accumulation over 150+ frames.
    *   **New Metrics and User Studies:** We have supplemented our FVD results with the **VMMD perceptual metric** and **user studies**, both of which corroborate MiMo's improvements in generation quality, especially in motion and condition following.

3.  **Clarified Contribution and Synergy with VFM-based Methods:** To clarify MiMo's novelty and its relation to methods like REPA (iS7b, 5Lni, xwwk), we have made two key updates:
    *   We position MiMo as a state-of-the-art **VFM-free** approach, highlighting its value as a self-contained solution.
    *   Crucially, we conducted a new experiment **combining MiMo with REPA**. The results show that the two methods are **complementary** and achieve state-of-the-art performance together. This key finding demonstrates that MiMo is a valuable and orthogonal contribution that can enhance both VFM-free and VFM-based VideoAR frameworks.

4.  **Addressed Practical Concerns:** We have included a detailed sensitivity analysis for key hyperparameters (masking ratio, loss weight $\lambda$) and a clear breakdown of the computational cost, showing that MiMo adds only a minimal (~10%) wall-clock time overhead during training for its substantial performance gains. We have also added a comprehensive implementation appendix to ensure full reproducibility.

We believe these comprehensive updates directly address the reviewers' concerns and substantially strengthen our paper. We are confident that our work makes a valuable contribution by identifying the underexplored role of history representations in VideoAR and offering a simple, efficient, and effective solution. We thank the reviewers again for their guidance.

---

### Comment · Area_Chair_phtx · 2025-11-27

Dear Reviewers,

This is a gentle reminder to please take a moment to review the author rebuttals and check whether your main concerns have been adequately addressed.

If possible, please update your reviews or add a brief clarification on whether the responses resolved your questions or if any issues remain. Your follow-up feedback is important for ensuring a fair and well-informed decision process.

Thank you again for your time and for helping maintain the quality of the ICLR review process.

Best,
AC

---

### Meta-Review · Area_Chair_FQSS · 2026-01-07

**Summary:**

The paper proposes MiMo, an approach to improve autoregressive video generation with representation learning. More specifically, the paper applies a mask modeling strategy during training to obtain a better representation of history frames.

Major concerns from the reviewers are (1) Can MiMo further improve the model with REPA already applied?  (2) The model is only validated on simple datasets. The rebuttal has addressed these concerns. Therefore, I would recommend acceptance of this work. I encourage the authors to incorporate reviewers' suggestions in their next version.

**Reviewer Concerns:**

Most of the major concerns are addressed by the rebuttal.

**Reviewer Scores:**

6, 8, 4, 4 -> 6, 8, 6, 6

---

### Decision · Program_Chairs · 2026-01-26

Accept (Poster)